# MORF and MOZ acetyltransferases target unmethylated CpG islands through the winged helix domain

Dustin C. Becht [1,10], Brianna J. Klein [1,10], Akinori Kanai [2,10], Suk Min Jang[3,10], Khan L. Cox[4], Bing-Rui Zhou [5], Sabrina K. Phanor[6], Yi Zhang [1], Ruo-Wen Chen[4], Christopher C. Ebmeier [7], Catherine Lachance[3], Maxime Galloy[3], Amelie Fradet-Turcotte [3], Martha L. Bulyk [6,8], Yawen Bai [5], Michael G. Poirier[4], Jacques Côté [3]✉, Akihiko Yokoyama [9]✉ & Tatiana G. Kutateladze [1]✉

Human acetyltransferases MOZ and MORF are implicated in chromosomal translocations associated with aggressive leukemias. Oncogenic translocations involve the far amino terminus of MOZ/MORF, the function of which remains unclear. Here, we identified and characterized two structured winged helix (WH) domains, WH1 and WH2, in MORF and MOZ. WHs bind DNA in a cooperative manner, with WH1 specifically recognizing unmethylated CpG sequences. Structural and genomic analyses show that the DNA binding function of WHs targets MORF/MOZ to gene promoters, stimulating transcription and H3K23 acetylation, and WH1 recruits oncogenic fusions to *HOXA* genes that trigger leukemogenesis. Cryo-EM, NMR, mass spectrometry and mutagenesis studies provide mechanistic insight into the DNA-binding mechanism, which includes the association of WH1 with the CpG-containing linker DNA and binding of WH2 to the dyad of the nucleosome. The discovery of WHs in MORF and MOZ and their DNA binding functions could open an avenue in developing therapeutics to treat diseases associated with aberrant MOZ/MORF acetyltransferase activities.

Fundamental processes in eukaryotic cells are commonly regulated through covalent modifications of DNA and posttranslational modifications (PTMs) of proteins. One of the canonical PTMs associated with transcriptionally active chromatin is acetylation of lysine residues of histones[1,2]. Acetylation removes the positive charge from the lysine side chain, weakening electrostatic contacts between histones and DNA, relaxing chromatin, and making DNA more accessible. Acetyllysine also serves as a docking site for numerous proteins and

[1]Department of Pharmacology, University of Colorado School of Medicine, Aurora, CO 80045, USA. [2]Department of Computational Biology and Medical Sciences, Graduate School of Frontier Sciences, the University of Tokyo, Kashiwa, Chiba 277-0882, Japan. [3]Laval University Cancer Research Center, CHU de Québec-UL Research Center-Oncology Division, Quebec City, QC G1R 3S3, Canada. [4]Department of Physics, Ohio State University, Columbus, OH 43210, USA. [5]Laboratory of Biochemistry and Molecular Biology, National Cancer Institute, National Institutes of Health, Bethesda, MD 20892, USA. [6]Division of Genetics, Department of Medicine, Brigham and Women's Hospital and Harvard Medical School, Boston, MA 02115, USA. [7]Department of Biochemistry, University of Colorado, Boulder, CO 80303, USA. [8]Department of Pathology, Brigham and Women's Hospital and Harvard Medical School, Boston, MA 02115, USA. [9]Tsuruoka Metabolomics Laboratory, National Cancer Center, Tsuruoka, Yamagata 997-0052, Japan. [10]These authors contributed equally: Dustin C. Becht, Brianna J. Klein, Akinori Kanai, Suk Min Jang. ✉e-mail: jacques.cote@crchudequebec.ulaval.ca; ayokoyam@ncc-tmc.jp; tatiana.kutateladze@cuanschutz.edu

complexes essential in gene transcription and DNA damage repair[3]. In mammals, acetylation is catalyzed by lysine acetyltransferase (KAT) complexes, including the MYST (Moz, Ybf2/Sas3, Sas2, Tip60) family of KATs. Among five members of the MYST family are the MOZ (monocytic leukemic zinc-finger protein) complex and the MORF (MOZ-related factor) complex[4,5]. The MOZ/MORF complexes play critical roles in embryogenesis, development, hematopoiesis, skeletogenesis, and cellular senescence and are involved in chromosomal translocations known to induce aggressive forms of blood cancer[6–14]. Acute leukemias derived from oncogenic MOZ/MORF translocations and aberrant acetyltransferase activities are associated with poor prognosis and grim survival rates, prompting and accelerating the development of inhibitors for MOZ/MORF with several already showing promising results as anti-cancer therapeutics[15,16]. Pathogenic MOZ/MORF have also been linked to developmental disorders, epilepsy, and intellectual disability[17–20].

The MOZ/MORF complexes acetylate primarily lysine 23 of histone H3 (H3K23ac) and contain four subunits[7,21,22]. A bromodomain PHD finger protein 1 (BRPF1) forms a scaffold for the assembly of other three subunits−the catalytic subunit MOZ/MORF, also known as KAT6A/KAT6B, inhibitor of growth 4/5 (ING4/5), and MYST/Esa1-associated factor 6 (MEAF6). The catalytic MOZ/MORF subunits are large, 2004/2073-amino acid proteins characterized by similar domain architecture. Both contain a double PHD finger (DPF) that recognizes acylated lysine 14 of histone H3 (H3K14acyl), the catalytic MYST domain, and the ED (glutamate/aspartate-rich) and SM (serine/methionine-rich) regions that were proposed to have a role in transcriptional activation[23–29] (Fig. 1a). Genetic and biochemical studies have shown that binding of the DPF domain to H3K14ac contributes to chromatin targeting by MOZ/MORF[23,24]. It stimulates H3K23 acetylation, activating gene transcription, and there is a positive crosstalk between H3K23ac and H3K14ac at the genomic sites occupied by MORF[21]. The functional importance of other regions of MOZ/MORF, beyond the DPF and MYST domains, particularly their N-termini, remains unclear.

In this study, we identified and characterized the tandem winged helix (WH) domains of MORF (MORF$_{WH1}$ and MORF$_{WH2}$) and MOZ (MOZ$_{WH1}$ and MOZ$_{WH2}$). We show that both WHs bind DNA but select for distinctive sequences, with MORF/MOZ$_{WH1}$ being highly specific toward unmethylated CpG. DNA binding function of WHs is required for the recruitment of MORF/MOZ to target gene promoters and H3K23 acetylation. Together, our structural, biochemical and in vivo findings reveal a previously uncharacterized mechanism by which a tandem of WH domains binds to the nucleosome, mediating the association of the major human acetyltransferases with specific genomic loci and their enzymatic functions.

## Results and discussion

### MORF and MOZ contain two DNA-binding winged helix (WH) domains

We have previously shown that DPF of MORF (MORF$_{DPF}$) associates with H3K14acyl and DNA, however, its low µM binding affinity suggests that this domain is not a major driver for the recruitment of the large MORF protein to chromatin. Searching for uncharacterized regions of MORF that could contribute to binding to chromatin, we explored the N-terminus of MORF (aa 1–182 of MORF, MORF$_{182}$). Dispersion of amide resonances in $^1$H,$^{15}$N heteronuclear single quantum coherence (HSQC) NMR spectrum of $^{15}$N-labeled MORF$_{182}$ indicated that this region is folded (Fig. 1b). Shorter constructs, generated by splitting MORF$_{182}$ in two halves, retained the fold, and their $^1$H,$^{15}$N HSQC spectra overlaid very well with the $^1$H,$^{15}$N HSQC spectrum of MORF$_{182}$. These results suggest that MORF$_{182}$ is comprised of two independent folded domains that have similar chemical environments either as linked or isolated modules and therefore likely do not interact with each other. We identified the first half of MORF$_{182}$ as a winged helix 1 (MORF$_{WH1}$)

and the second half as a winged helix 2 (MORF$_{WH2}$) based on the data described below, and from here on refer to them as MORF$_{WHs}$.

To determine whether MORF$_{WHs}$ are capable of binding to DNA, we examined the association of MORF$_{WH1}$ and MORF$_{WH2}$ with 147 bp 601 Widom DNA (DNA$_{147}$) in an electrophoretic mobility shift assay (EMSA). DNA$_{147}$ was incubated with increasing amounts of MORF$_{WH1}$ and MORF$_{WH2}$ and the reaction mixtures were resolved on native polyacrylamide gels (Fig. 1c, d). A gradual increase in the amounts of added MORF$_{WH1}$ and MORF$_{WH2}$ caused a shift of the DNA$_{147}$ band and the appearance of several bands corresponding to the complexes formed between MORF$_{WH1}$ and MORF$_{WH2}$ and multiple major/minor grooves[30] of DNA$_{147}$. The binding to DNA$_{147}$ was confirmed by NMR titration experiments. Upon addition of DNA$_{147}$, amide crosspeaks of MORF$_{WH2}$ broadened beyond detection due to the formation of large MORF$_{WH2}$-DNA$_{147}$ complexes (Supplementary Fig. 1).

A high degree similarity of amino acid sequences between MORF and homologous MOZ suggested that MOZ also contains two N-terminal WHs, MOZ$_{WH1}$, and MOZ$_{WH2}$ (Fig. 1e). Indeed, the dispersion of amide resonances in $^1$H,$^{15}$N HSQC spectra of MOZ$_{WH1}$ and MOZ$_{WH2}$ pointed to the presence of independent folded modules (Supplementary Fig. 2a). Both MOZ$_{WH1}$ and MOZ$_{WH2}$ readily shifted the DNA$_{147}$ band in EMSA, confirming that the DNA binding activity is conserved in MORF and MOZ (Supplementary Fig. 2b).

### MORF$_{WHs}$ are required for MORF recruitment to chromatin and H3K23 acetylation

Are MORF$_{WHs}$ essential for biological functions of MORF? We investigated the role of MORF$_{WHs}$ in genomic occupancy of MORF and MORF-dependent H3K23 acetylation in vivo by chromatin immunoprecipitation (ChIP) experiments (Fig. 1f–h and Supplementary Fig. 3a). Human K562 cells expressing FLAG-tagged MORF$_N$ (aa 1–716 of MORF, containing MORF$_{WH1}$, MORF$_{WH2}$, MORF$_{DPF}$ and MORF$_{MYST}$), wild type (WT) or mutants in which MORF$_{WH1}$ (MORF$_N$ ΔWH1), MORF$_{WH2}$ (MORF$_N$ ΔWH2) or both MORF$_{WHs}$ (MORF$_N$ ΔWH1-WH2) are deleted, were generated and used to measure MORF$_N$, H3K23ac and H3K14ac levels at promoters of a set of target genes. Compared to the binding of WT MORF$_N$, loss of MORF$_{WH1}$ substantially decreased the binding of MORF$_N$ to promoters of all genes tested, and while loss of MORF$_{WH2}$ had a milder impact, deletion of both MORF$_{WHs}$ had a cumulative effect in disrupting the recruitment of MORF$_N$ to chromatin (Fig. 1f). Furthermore, expression of MORF$_N$ with both MORF$_{WHs}$ being deleted led to a decrease in the level of H3K23 acetylation at these promoters compared to the H3K23ac level observed upon expression of WT MORF$_N$, with the loss of MORF$_{WH1}$ resulting in a more notable change than the loss of MORF$_{WH2}$ (Fig. 1g). As expected, H3K14ac level was reduced by the deletion of MORF$_{WHs}$ to a lesser extent than the H3K23ac level and was essentially unaffected by the deletion of MORF$_{WH2}$ (Fig. 1h). Together, these data demonstrate that both functional MORF$_{WH1}$ and MORF$_{WH2}$ are required for MORF to occupy its target genes and acetylate H3K23 in vivo.

### MORF$_{WH2}$ binds DNA via its α3 and α2 helices

To gain insight into the DNA binding mechanisms of MORF$_{WHs}$, we assessed the minimal size of DNA to which these domains can bind. EMSA experiments using increasing amounts of MORF$_{WH1}$ and MORF$_{WH2}$ and a 10 bp to 100 bp DNA ladder revealed that either domain interacts with a double-stranded DNA at least or larger than 15 bp in length (Supplementary Fig. 4). In support, GST-MORF$_{WH2}$ immobilized onto glutathione sepharose beads pulled down fluorescein (FAM)-labeled 37 bp dsDNA (FAM-DNA$_{37}$) in a confocal microscopy assay (Fig. 2a), and a 15 bp A-rich dsDNA (A-DNA$_{15}$) induced chemical shift perturbations (CSPs) in $^{15}$N-labeled MORF$_{WH2}$ in $^1$H,$^{15}$N HSQC titration experiments (Fig. 2b). To identify residues of MORF$_{WH2}$ responsible for binding to DNA, we collected and analyzed triple resonance NMR spectra of uniformly $^{13}$C,$^{15}$N-labeled MORF$_{WH2}$ and assigned backbone

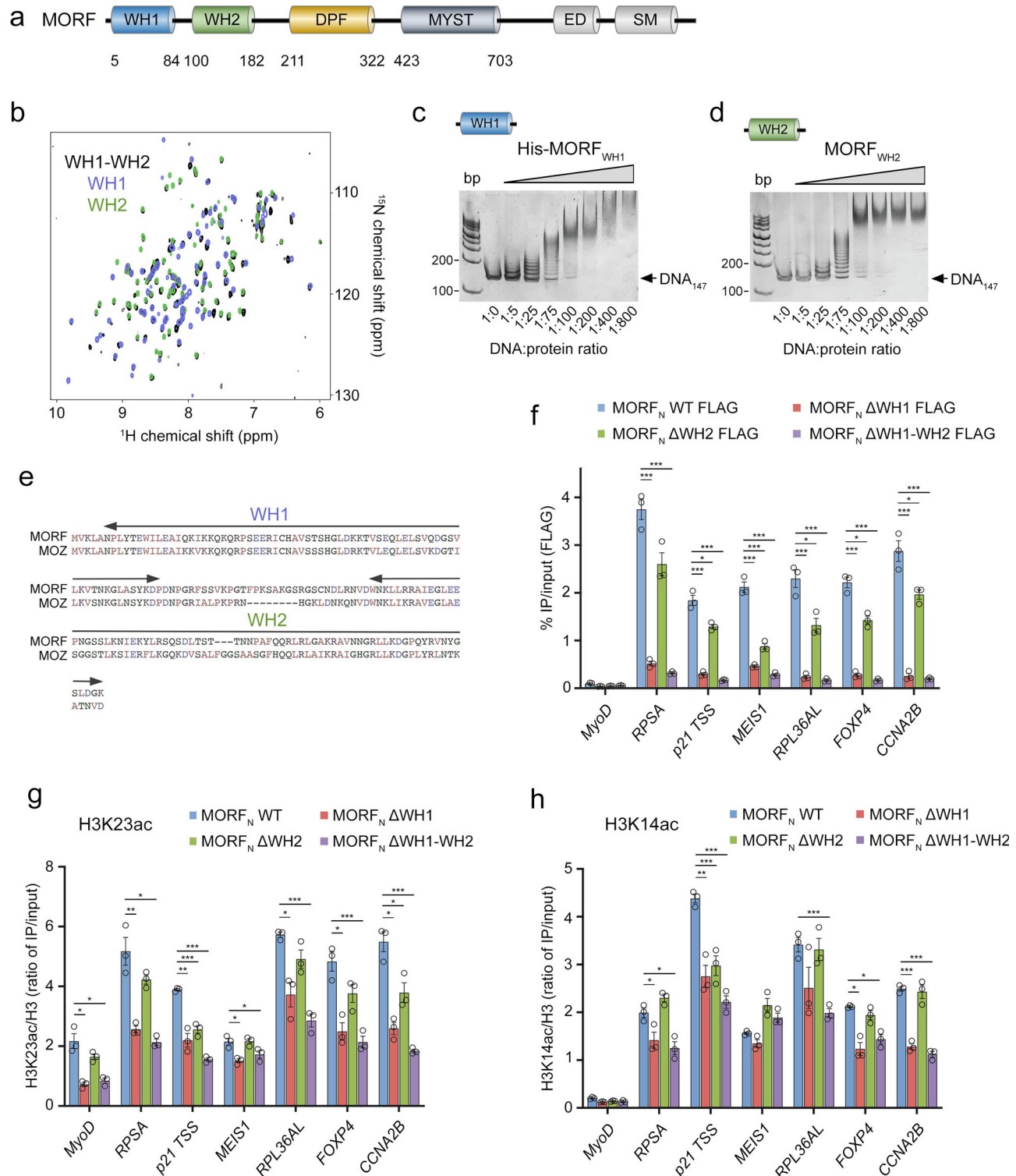

**Fig. 1 | MORF and MOZ contain two DNA-binding WH domains. a** Domain architecture of MORF with the MORF$_{WH1}$, MORF$_{WH2}$ and MORF$_{DPF}$ colored blue, green and wheat, respectively. **b** Overlay of $^1$H,$^{15}$N HSQC spectra of $^{15}$N-labeled MORF$_{WH1-WH2}$ (black), MORF$_{WH1}$ (blue) and MORF$_{WH2}$ (green). **c, d** EMSA of 147 bp 601 DNA in the presence of increasing amounts of His-MORF$_{WH1}$ and MORF$_{WH2}$. DNA:protein ratio is shown below the gel images. **e** Alignment of amino acid sequences of the N-terminal regions (1–200) of MORF and MOZ. Boundaries of WH1 and WH2 are indicated by arrows. **f–h** Anti-FLAG ChIP-qPCR analysis in K562 cells

stably expressing WT, ΔWH1, ΔWH2, and ΔWH1-WH2 FLAG-MORF$_{N (1-716)}$, presented as IP/input % at the indicated genes (near TSS). ChIP analysis of H3K23ac (**g**) and H3K14ac (**h**) in K562 cells as in (**f**). Acetylation levels were corrected for nucleosome occupancy (total H3 signal), presented as a ratio of IP/input (H3ac/total H3). Data represent mean ± SEM from three independent experiments. n = 3 Statistical tests were two-sided. Student's *t*-test, ***$P < 0.005$, $0.005 < **P < 0.01$, and $0.01 < *P < 0.05$. Source data are provided as a Source Data file.

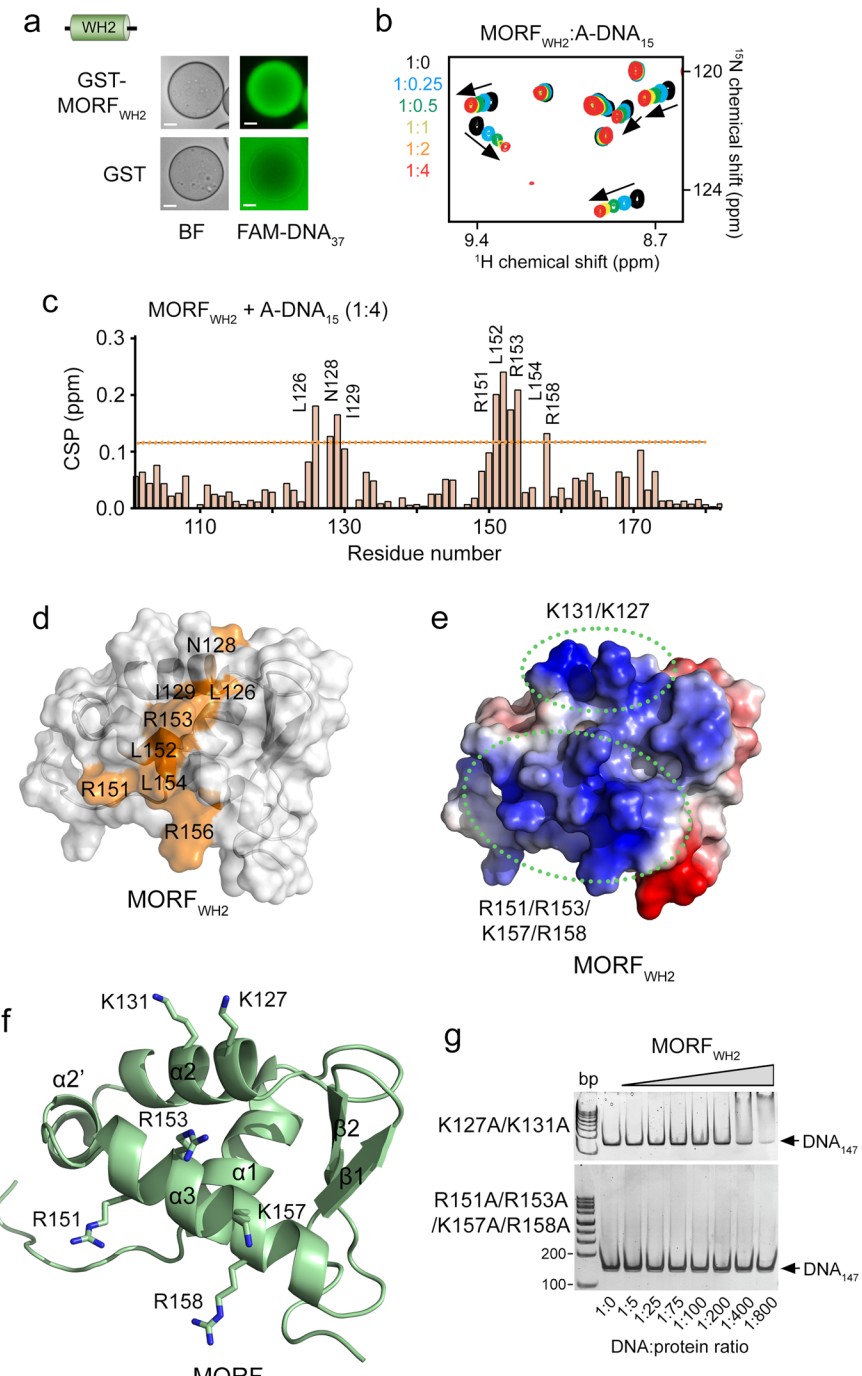

**Fig. 2 | Structural basis for binding of MORFWH2 to DNA. a** Confocal images of GST-MORF$_{WH2}$ or GST (control) bound to glutathione Sepharose beads in the presence of FAM-labeled 37 bp dsDNA. Scale bar, 20 μm. **b** Overlay of $^1$H,$^{15}$N HSQC spectra of MORF$_{WH2}$ in the presence of increasing amount of 15 bp A-rich dsDNA (A-DNA$_{15}$). Spectra are color coded according to the protein:DNA molar ratio. **c** Histogram of normalized CSPs in $^1$H,$^{15}$N HSQC spectra of MORF$_{WH2}$ induced by a fourfold molar excess of A-DNA$_{15}$ as a function of residue. **d–f** The solution NMR structure of MORF$_{WH2}$. Residues of MORF$_{WH2}$ that exhibited large CSPs upon addition of A-DNA$_{15}$ (greater than average plus one standard deviation, indicated by the dotted line in **c**) are mapped on the structure of MORF$_{WH2}$, colored orange and

labeled in **d**. Electrostatic surface potential of MORF$_{WH2}$, with blue and red colors representing positive and negative charges, respectively, is shown in **e**. Two positively charged regions of MORF$_{WH2}$ are indicated by green dotted ovals, with lysine and arginine residues labeled. Ribbon diagram of the MORF$_{WH2}$ structure in the same orientation as in **d** and **e** is shown in **f**. Lysine and arginine residues of the two positively charged regions are shown in sticks and labeled. **g** EMSA of 147 bp 601 DNA in the presence of increasing amounts of K127A/K131A MORF$_{WH2}$ and R151A/R153A/K157A/R158A MORF$_{WH2}$ mutants. DNA:protein ratio is shown below the gel images. Source data are provided as a Source Data file.

amide resonances. In solution, A-DNA$_{15}$ induced CSPs primarily in two regions, encompassing residues L126-I129 and R151-R158 of MORF$_{WH2}$ (Fig. 2c). We then determined the three-dimensional solution NMR structure of MORF$_{WH2}$ and mapped the most perturbed residues onto the structure (Fig. 2d). The structure revealed a winged helix fold

consisting of three α-helices, a double-stranded β-sheet, one short wing connecting two β-strands, and an additional α2'-helix is present between α2 and α3 (Fig. 2d–f and Supplementary Table 1).

Analysis of electrostatic surface potential of MORF$_{WH2}$ showed that the most perturbed residues of MORF$_{WH2}$ are located in the highly

positively charged patch of the domain that could electrostatically interact with the negatively charged DNA (Fig. 2e). We mutated K127 and K131 in the α2 helix and separately R151, R153, K157 and R158 in the α3 helix to alanine and tested these mutants in EMSA (Fig. 2f, g). While the K127A/K131A MORF$_{WH2}$ mutant retained a weak DNA binding ability, binding of the R151A/R153A/K157A/R158A MORF$_{WH2}$ mutant to DNA$_{147}$ was abolished, indicating that both α2 and α3 are necessary for the strong interaction with DNA, with the α3 helix of MORF$_{WH2}$ being critical. We note that R151, K157, and R158 of MORF$_{WH2}$ are found mutated in adenocarcinoma, malignant melanoma and breast and stomach cancers (Cosmic).

## MORF$_{WH1}$ selects for CpG-rich DNA

To assess whether MORF$_{WHs}$ recognize specific DNA sequences, we tested them in universal oligonucleotide arrays that contain all possible combinations of 10 bp sequences within ~44,000 60-bp probes[31,32]. As shown in Fig. 3a, MORF$_{WH2}$ has a slight preference for an AT-rich DNA sequence. This minor selectivity of MORF$_{WH2}$ was confirmed by EMSA with A-DNA$_{15}$ and a 15 bp C-rich dsDNA (C-DNA$_{15}$) (Fig. 3b and Supplementary Fig. 4). Quantitative analysis of EMSAs yielded a 7.8 μM binding affinity of MORF$_{WH2}$ to A-DNA$_{15}$ and a two-fold weaker binding affinity of MORF$_{WH2}$ to C-DNA$_{15}$ ($K_d$=18 μM) (Fig. 3c, d). Methylation of cytosine did not affect binding, as MORF$_{WH2}$ associated with unmethylated 16 bp CpG dsDNA (CpG-DNA$_{16}$) and methylated CpG-DNA$_{16}$ (mCpG-DNA$_{16}$) equally well (Fig. 3b).

In contrast, MORF$_{WH1}$ bound specifically to the CpG-rich DNA sequences and did not recognize the AT-rich sequences in universal oligonucleotide arrays and EMSA (Fig. 3e, f). The dissociation constant for the interaction of MORF$_{WH1}$ with CpG-DNA$_{16}$ was found to be 2 μM, however this interaction was substantially diminished by methylation of the CpG motif (Fig. 3f, g). A computational model of MORF$_{WH1}$ suggested a winged three-helix fold; however the α3 helix, which contains a set of DNA-interacting lysine and arginine residues in MORF$_{WH2}$, does not contain the positively charged residues in MORF$_{WH1}$ and therefore likely does not mediate binding of MORF$_{WH1}$ to DNA. Instead, electrostatic surface potential of this model shows two positively charged regions encompassing the α1 helix, the loop connecting α1 and α2 and the loop connecting two β-strands in the β-hairpin (Fig. 3h). The model of MORF$_{WH1}$ superimposes with the crystal structure of WH from SAMD1 (rmsd of 0.6 Å) (Fig. 3i), an atypical WH that binds to DNA by a mechanism distinctly different from that of typical WHs[33]. While the C-terminal end of α1 helix and the loop connecting α1 and α2 in SAMD1 WH insert in the CpG-containing major groove of DNA, the atypically long β-hairpin inserts into a neighboring minor groove of DNA[33]. The overlay with the SAMD1 WH structure suggested that MORF$_{WH1}$ has a similar mode of binding to DNA (Fig. 3i). To test this, we generated the K19A/K21A/K22A mutant of MORF$_{WH1}$, harboring mutations in α1, and the K24A/R26A/K66A mutant of MORF$_{WH1}$, harboring mutations in the loops, and evaluated binding of these mutants to DNA$_{147}$ by EMSA (Fig. 3j, k). We found that mutation of K24, R26, and K66 completely disrupts binding of MORF$_{WH1}$ to DNA$_{147}$ and mutation of K19, K21 and K22 substantially decreases this interaction. In agreement, NMR titration experiments showed that wild-type MORF$_{WH1}$ tightly binds to CpG-DNA$_{16}$, exhibiting CSPs in the intermediate exchange regime on the NMR time scale (Fig. 3l and Supplementary Fig. 5), however the ability to bind CpG-DNA$_{16}$ was lost by the K24A/R26A/K66A mutant of MORF$_{WH1}$ or notably reduced by the K19A/K21A/K22A mutant of MORF$_{WH1}$ (Fig. 3m, n and Supplementary Fig. 5). Collectively, NMR and EMSA results suggest that similar to the DNA binding mode of atypical WH from SAMD1, the two loops and α1 of MORF$_{WH1}$ mediate binding to DNA and that the DNA binding mechanism of MORF$_{WH1}$ differs from the DNA binding mechanism of the typical MORF$_{WH2}$.

## MOZ$_{WH1}$ targets CpG genome wide

Analysis of DNA binding selectivity of MOZ$_{WH1}$ and MOZ$_{WH2}$ in universal oligonucleotide PBM arrays revealed that, similar to MORF$_{WHs}$, MOZ$_{WH1}$ specifically binds to the CpG-rich DNA sequences and MOZ$_{WH2}$ shows essentially no sequence selectivity (Supplementary Fig. 2c). We examined genomic localization of endogenous full-length MOZ (MOZ$_{FL}$) and a series of exogenously expressed FLAG-tagged shorter MOZ constructs, including MOZ$_{WH1}$, MOZ$_{WH2-DPF}$, and MOZ$_{WH1-WH2-DPF}$ in human HEK293T cells by chromatin immunoprecipitation coupled with deep sequencing (ChIP-seq). Analysis of ChIP-seq showed that the genome-wide occupancy of MOZ$_{WH1}$ centered around the transcription start sites (TSS) and correlated well with the distribution of unmethylated CpG, which was identified by CpG island recovery assay for unmethylated CpGs coupled with deep sequencing (CIRA-seq), as well as with non-phosphorylated RNAP2 (Fig. 4a, b). The ChIP signal intensities of MOZ$_{WH1}$ correlated with the signal intensities of MOZ$_{FL}$ ($r = 0.92$) and unmethylated CpG ($r = 0.86$) (Fig. 4c, d). In HEK293T cells, ~89% (11,021 out of 12,386) of the MOZ$_{FL}$-bound genes overlapped with ~97% (11,021 out of 11,381) of the MOZ$_{WH1}$-bound genes, and ~87% (10,840 out of 12,587) of the unmethylated CpG-enriched genes overlapped with ~96% (10,840 out of 11,381) of the MOZ$_{WH1}$-bound genes (Fig. 4e). Loss of MOZ$_{WH1}$ resulted in a relatively non-specific chromatin association of MOZ$_{WH2-DPF}$ (Fig. 4a, b) and only ~14% of the MOZ$_{FL}$-bound genes and ~12% of the unmethylated CpG-enriched genes were co-occupied by MOZ$_{WH2-DPF}$ (Fig. 4f). No correlation was observed between the ChIP/CIRA signals of MOZ$_{WH2-DPF}$ and MOZ$_{FL}$ ($r = -0.06$) and unmethylated CpG ($r = -013$) (Fig. 4g, h). Co-occupancy with unmethylated CpG sites however was restored for MOZ$_{WH1-WH2-DPF}$ (Fig. 4a, b). The high degree correlation between the ChIP/CIRA signals of MOZ$_{WH1-WH2-DPF}$ and MOZ$_{FL}$ ($r = 0.87$) and unmethylated CpG ($r = 0.84$) mirrored the correlation between MOZ$_{FL}$ and unmethylated CpG ($r = 0.88$) (Fig. 4i–k). We noticed a moderate enrichment of MOZ$_{FL}$ and MOZ$_{WH2-DPF}$ downstream of TSS (Fig. 4a, b). The ChIP signal intensity of H3K14ac also increased downstream of TSS, suggesting that binding of MOZ$_{DPF}$ to H3K14ac in transcribed regions contributes to the association of MOZ with chromatin (Fig. 4a, b). Together, the ChIP-seq and CIRA-seq data demonstrate that MOZ$_{WH1}$ targets MOZ to unmethylated CpG-rich regions, whereas the MOZ$_{DPF}$-H3K14ac interaction can provide additional anchoring in transcribed regions.

## MOZ/MORF$_{WH1}$ is required for binding of MOZ/MORF to target genes

The positive correlation between MOZ$_{WH1}$ or MOZ$_{WH1-WH2-DPF}$ levels and unmethylated CpG levels at TSS was also observed at individual genes, such as the HOXA family, MYC and CDKN2C (Fig. 5a). The distribution pattern of MOZ$_{WH1}$ and MOZ$_{WH1-WH2-DPF}$ was similar to that of MOZ$_{FL}$ and other components of the MOZ complex, ING4 and MEAF6, and replicated the distribution of unmethylated CpG in gene promoters, including promoters of oncogenic HOXA9 and MYC, known to induce leukemogenesis. Loss of MOZ$_{WH1}$ led to a non-specific binding of MOZ$_{WH2-DPF}$ throughout the genomic regions tested, supporting the notion that the major determinant of genomic occupancy of MOZ is MOZ$_{WH1}$, which binds unmethylated CpG-rich promoters.

Occupancy of MORF$_{WH1}$ at the promoter regions mirrored occupancy of MOZ$_{WH1}$ (Fig. 5b). The localization patterns of MORF$_{WH1}$ and MOZ$_{WH1}$ at the leukemic oncogenes HOXA9 and MYC were nearly identical (Fig. 5c). ChIP-qPCR analysis confirmed that MORF$_{WH1}$ specifically occupies the promoters of MYC and HOXA9, whereas the K19A/K21A/K22A and K24A/R26A/K66A mutants of MORF$_{WH1}$, defective in unmethylated CpG binding (Fig. 3 j, k, m, n) were unable to localize to the specific genomic sites (Fig. 5d). These results indicate that recognition of the CpG sequence by WH1 is conserved in MORF and MOZ and is required for binding to chromatin in vivo.

One of the chromosomal translocations directly linked to the development of acute myeloid leukemia (AML) is the fusion of MOZ

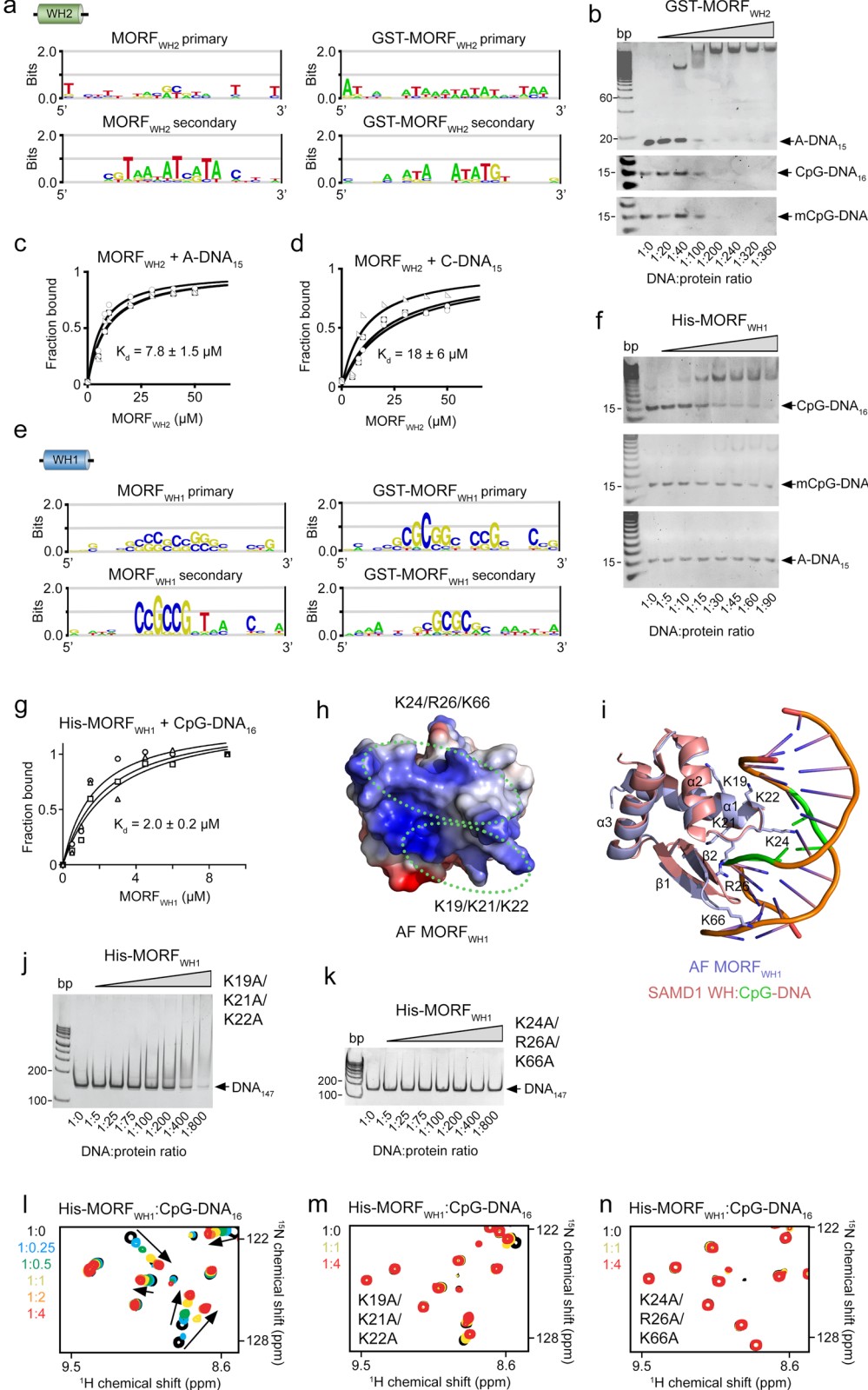

with a transcriptional co-activator TIF2 (Fig. 6a), which in turn associates with another co-activator, CPB/p300[34–36]. As shown in Fig. 5a, MOZ-TIF2 co-localized with endogenous $MOZ_{FL}$, unmethylated CpGs, $MOZ_{WH1}$ and $MOZ_{WH1-WH2-DPF}$ at the same gene promoters, indicating that $MOZ_{WH1}$ drives the recruitment of the leukemogenic MOZ-TIF2 chimera to chromatin. To further characterize this recruitment, we tested occupancy of $MOZ_{FL}$, MOZ-TIF2 and MOZ-TIF2 ΔWH1 at the

target genes *MYC*, *HOXA9*, and *CDKN2C*, as well as negative control *CD4*, by ChIP-qPCR (Fig. 6b and Supplementary Fig. 3b). $MOZ_{FL}$ and MOZ-TIF2 showed similar chromatin binding patterns within all three regions tested, whereas the deletion of $MOZ_{WH1}$ abolished chromatin binding activity of MOZ-TIF2 ΔWH1 (Fig. 6b). Recruitment of the MORF-TIF2 fusion to chromatin also depended on $MORF_{WH1}$ because MORF-TIF2 ΔWH1 was unable to bind to the *MYC, HOXA9,* and *CDKN2C*

**Fig. 3 | DNA sequence selectivity of MORFWHs. a** Analysis of DNA binding selectivity of $MORF_{WH2}$ in universal oligonucleotide PBMs. One of the replicates met significance criteria. **b** EMSAs of 15 bp A-rich, 16 bp unmethylated CpG, and 16 bp methylated CpG dsDNA in the presence of increasing amounts of GST-$MORF_{WH2}$. DNA:protein ratio is shown below the gel images. **c, d** EMSA-derived binding curves used to determine $K_d$ values for the interaction of $MORF_{WH2}$ with the indicated DNA. Data represent average ± SD of three independent experiments. $n = 3$. **e** Analysis of DNA binding specificity of $MORF_{WH1}$ in universal oligonucleotide PBMs. **f** EMSAs of 16 bp unmethylated CpG, 16 bp methylated CpG, and 15 bp A-rich dsDNA in the presence of increasing amounts of His-$MORF_{WH1}$. DNA:protein ratio is shown below the gel images. **g** EMSA-derived binding curves used to determine $K_d$ values for the interaction of His-$MORF_{WH1}$ with 16 bp unmethylated CpG dsDNA. Data represent average ± SD of four independent experiments. $n = 4$. **h** Electrostatic

surface potential of the alpha fold (AF) computational model of $MORF_{WH1}$ from UniProt (Q8WYB5), with blue and red colors representing positive and negative charges, respectively. Two positively charged regions of $MORF_{WH1}$ are indicated by green dotted ovals, with lysine and arginine residues labeled. **i** Superimposition of the AF model of $MORF_{WH1}$ from UniProt (Q8WYB5) (blue) and the crystal structure of the DNA-bound WH from SAMD1 (salmon), with CpG colored green (PDB ID 6LUI). The mutated residues of $MORF_{WH1}$ are shown as sticks and labeled. **j, k** EMSA of 147 bp 601 DNA in the presence of increasing amounts of K19A/K21A/K22A $MORF_{WH1}$ (**j**) and K24A/R26A/K66A $MORF_{WH1}$ (**k**) mutants. DNA:protein ratio is shown below the gel images. **l–n** Overlay of $^1H,^{15}N$ HSQC spectra of His-$MORF_{WH1}$, wild type (**l**), K19A/K21A/K22A (**m**) or K24A/R26A/K66A (**n**) in the presence of increasing amount of CpG-$DNA_{16}$. Spectra are color coded according to the protein:DNA molar ratio. Source data are provided as a Source Data file.

genes (Fig. 6c and Supplementary Fig. 3b). Mutations in $MORF_{WH2}$ that impair its DNA binding (R151A/R153A/K157A/R158A) reduced the gene-specific localization of MORF-TIF2 WH2-mutant compared to localization of MORF-TIF2. Much like MORF-TIF2 ΔWH1, MORF-TIF2 ΔWH1 + R151A/R153A/K157A/R158A mutant completely lost its ability to associate with *MYC, HOXA9* and *CDKN2C* genes (Fig. 6c), confirming that this region is indispensable for the retention of MOZ/MORF-TIF2 at chromatin.

### MOZ/MORF_WH1 is essential in leukemogenicity of MOZ/MORF-TIF2

To determine the role of WHs in leukemogenic activity of the MOZ/MORF-TIF2 fusions, we performed myeloid progenitor transformation assay, in which c-Kit positive hematopoietic progenitors were transduced with the MOZ fusions (MOZ-TIF2 and MOZ-TIF2 ΔWH1) and the MORF counterparts (MORF-TIF2, MORF-TIF2 ΔWH1, the MORF-TIF2 $MORF_{WH2}$ mutant, and MORF-TIF2 ΔWH1 + $MORF_{WH2}$ mutant) and cultured in a semi-solid media ex vivo (Fig. 6d). Consistent with their chromatin binding abilities (Fig. 6b, c), MORF-TIF2 ΔWH1 and MOZ-TIF2 ΔWH1, in which WH1 was deleted, failed to activate *Hoxa9* gene expression and immortalize hematopoietic progenitors, a characteristic feature of oncogenic MOZ fusions (Fig. 6e, f). These results indicate that WH1 is a critical chromatin targeting module for the MOZ and MORF fusions and is necessary for leukemogenesis. The inactivating mutation of $MORF_{WH2}$ modestly hampered Hoxa9 expression in the second passage in cells transduced with the MORF-TIF2 $MORF_{WH2}$ mutant compared to the cells transduced with wild-type MORF-TIF2 (Fig. 6e). The MORF-TIF2 $MORF_{WH2}$ mutant expressing cells gradually lost their clonogenicities and failed to form colonies in the fourth round passage, indicating that the intact $MORF_{WH2}$ is required for the full leukemogenicity of MORF-TIF2 (Fig. 6f). Collectively, these data demonstrate that the WH1-mediated binding to unmethylated CpG-rich DNA is crucial for the oncogenic activity of the MOZ/MORF fusions, and while WH2 also plays a role, it appears to be less drastic than that of WH1.

### Concomitant engagement of MORF_WHs augments binding to the nucleosome

To characterize the DNA binding mechanism of WHs in detail, we investigated the association of $MORF_{WH1}$ and $MORF_{WH2}$ with the nucleosome core particle (NCP) by EMSA and fluorescence anisotropy assays (Fig. 7a–i). For EMSA, we used nucleosomes containing a 147 bp 601 DNA ($NCP_{147}$) and a 187 bp 601 DNA ($NCP_{187}$) and for fluorescence anisotropy measurements, we reconstituted fluorescein-labeled $NCP_{147}$ and $NCP_{207}$. $NCP_{207}$ was generated using a 207 bp DNA in which 147 bp 601 DNA is flanked by 30 bp linker DNA on either side and internally labeled with fluorescein 27 bp in from the 5' end. Both $MORF_{WH1}$ and $MORF_{WH2}$ shifted the $NCP_{147}$ and $NCP_{187}$ bands in EMSA, indicating the formation of the $MORF_{WH1}$-NCP and $MORF_{WH2}$-NCP complexes (Fig. 7a, b, d, e). While the presence of an extra-nucleosomal linker DNA in $NCP_{187}$ increased the association of

$MORF_{WH1}$, binding of $MORF_{WH2}$ was unaffected. In support, quantitative measurements of binding affinities by fluorescence polarization revealed that $MORF_{WH2}$ does not discriminate between $NCP_{147}$ and $NCP_{187}$ and interacts equally well with either nucleosome ($S_{1/2} = 1.0\ \mu M$ and 1.4 μM, respectively) (Fig. 7c). These data suggest that $MORF_{WH2}$ utilizes the same mechanism for binding to the nucleosome regardless of the presence of extra-nucleosomal DNA fragments, still, the nucleosome organization is essential because binding of $MORF_{WH2}$ to the nucleosomes was ~4–6-fold tighter than its binding to $DNA_{147}$.

Titration of $MORF_{WH1}$ against $NCP_{147}$ yielded a $S_{1/2}$ of 9 μM for the $MORF_{WH1}$-$NCP_{147}$ complex formation, however $MORF_{WH1}$ associated 30-fold tighter with $NCP_{207}$ ($S_{1/2} = 0.3\ \mu M$), indicating its preference for a linear, free of the nucleosome DNA (Fig. 7f). Binding affinity of $MORF_{WH1}$ to $DNA_{147}$ ($S_{1/2} = 15\ \mu M$) was only slightly weaker compared to its binding affinity to $NCP_{147}$ ($S_{1/2} = 9\ \mu M$). The linked $MORF_{WH1-WH2}$ construct exhibited affinities of 7 nM to $NCP_{207}$, 12 nM to $NCP_{147}$, and 62 nM to $DNA_{147}$, which indicated a cooperative binding of two independent $MORF_{WHs}$ (Fig. 7i) and similar behavior was observed for $MOZ_{WH1-WH2}$ (Supplementary Fig. 6). The absence of CSPs in $^{15}N$-labeled $MORF_{WH2}$ upon addition of unlabeled $MORF_{WH1}$ confirmed that the two $MORF_{WHs}$ do not interact (Fig. 7j and Supplementary Fig. 7). In support of EMSA and NMR titration data (Fig. 3j–n), the K24A/R26A/K66A and K19A/K21A/K22A mutants of $MORF_{WH1}$ were essentially incapable of binding to $NCP_{147}$ (Fig. 7k).

### MORF_WH2 binds to the dyad of the nucleosome and MORF_WH1 binds to the CpG linker DNA

To define the structural basis for the association of $MORF_{WHs}$ with the nucleosome, we obtained a 7 Å resolution map of a 197 bp NCP ($NCP_{197}$) in complex with $MORF_{WH1-WH2}$ by cryo-electron microscopy (cryo-EM) (Fig. 8a, b). For reconstitution of $NCP_{197}$ we used $DNA_{197}$ in which 147 bp Widom 601 DNA is flanked by two linker DNA fragments. One linker contains three CpGs and another contains one CpG. The formation of the $MORF_{WH1-WH2}$-$NCP_{197}$ complex was monitored in EMSA. The cryo-EM map of the $MORF_{WH1-WH2}$-$NCP_{197}$-scFv complex showed extra density near the nucleosome dyad region. DNA and histones of the $NCP_{197}$ structure (PDB ID: 7K5X) and $MORF_{WH2}$ were readily docked into the cryo-EM density map (Fig. 8a, b and Supplementary Fig. 8). In addition to the density of $MORF_{WH2}$ at the NCP dyad, colored green in Fig. 8a, b, we observed weaker extra density on the linker DNA around the C22, G23 DNA sequence, colored blue. Excellent superimposition of the structure of the CpG-bound atypical SAMD1 WH with the CpG (C22 and G23) region of the cryo-EM structure suggested that this weaker extra density belongs to $MORF_{WH1}$ (Fig. 8c, d).

To verify the mechanism by which $MORF_{WH1}$ recognizes CpG-$NCP_{197}$, we tested wild-type $MORF_{WH1}$ and the impaired in binding to CpG-$DNA_{16}$ mutants K19A/K21A/K22A and K24A/R26A/K66A in EMSA (Fig. 8e and Supplementary Fig. 9). While wild type $MORF_{WH1}$ formed a complex with $NCP_{197}$, both mutants were unable to bind to the nucleosome. These results are in agreement with the NMR data

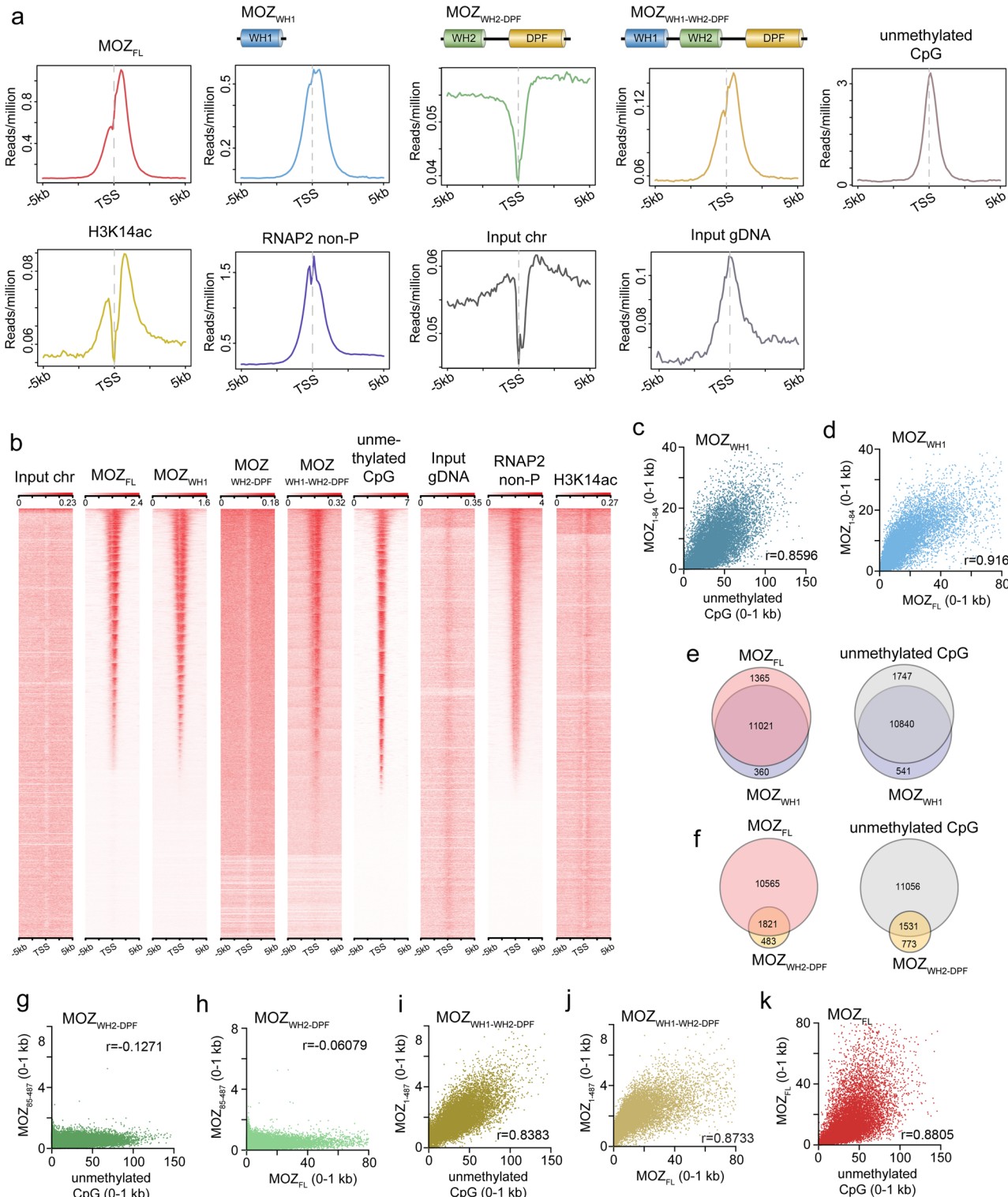

**Fig. 4 | MOZWH1 targets unmethylated CpG genome wide. a** Average distribution of various MOZ proteins near the TSSs. The ChIP signal distribution of endogenous full length MOZ (MOZ_FL), FLAG-tagged MOZ (MOZ_WH1, MOZ_WH2, MOZ_WH2-DPF, MOZ_WH1-WH2-DPF), endogenous H3K14ac, and RNA Polymerase II with the non-phosphorylated CTD heptapeptide motif (RNAP2 non-P) within the 5 kb range of TSS is shown using ngsplot. The distribution of unmethyated CpGs was determined by CpG island recovery assay coupled with deep sequencing (CIRA-seq). **b** Heatmaps of ChIP signal intensities of indicated proteins/modifications at TSS are shown using ngsplot. **c, d** Correlations between the ChIP/CIRA signal intensities of MOZ_WH1 and

unmethylated CpGs (**c**) or MOZ_FL (**d**). The ChIP/CIRA-seq tags were clustered into a 1 kb bin (0 to +1 kb from the TSS) and are presented as XY scatter plots with the Spearman's rank correlation coefficient (*r*). **e, f** Venn diagrams of the MOZ_WH1-occupied genes and MOZ_FL-occupied genes or unmethylated CpG-rich genes are shown. The ChIP/CIRA-seq tag counts in the 1 kb bin were divided by the corresponding input chromatin/DNA tag counts, and the genes whose relative values are larger than 2 were used for analysis. **g–k** Correlations between the ChIP/CIRA signal intensities of the indicated MOZ proteins and unmethylated CpGs. The data were analyzed as in **c, d**. Source data are provided as a Source Data file.

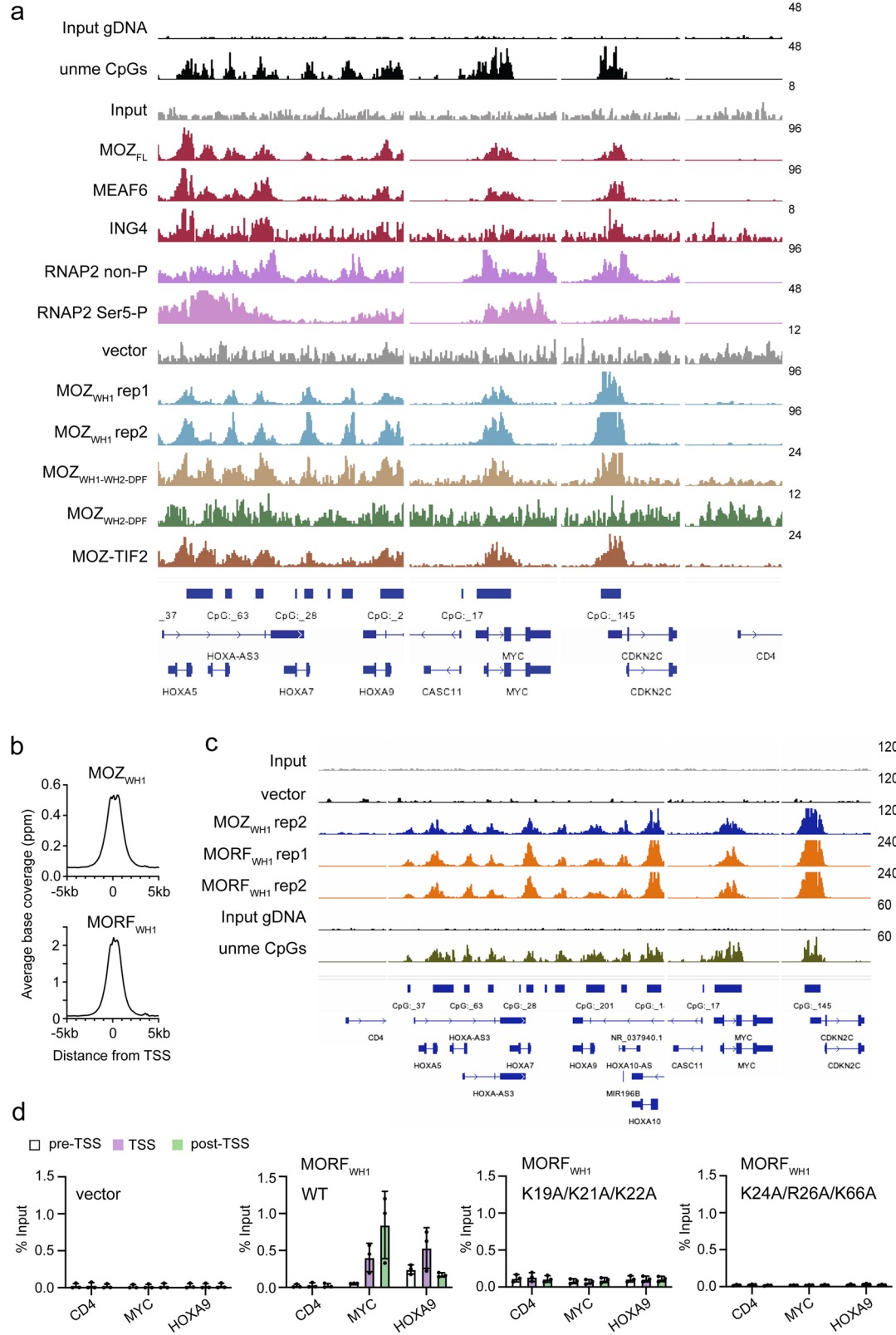

described above for the interaction of MORF$_{WH1}$ with CpG-DNA$_{16}$ (Fig. 3), reinforcing the idea that the atypical MORF$_{WH1}$ requires intact α1 and the two loops for its binding to CpG-NCP.

In contrast to MORF$_{WH1}$, MORF$_{WH2}$ is a typical WH that binds to DNA through its α3 and α2 helices[37]. The structure of MORF$_{WH2}$ aligns well (rmsd of 0.9 Å) with the structures of H1 and H5, the linker histones known to bind to the dyad of the nucleosome, decreasing

spontaneous DNA unwrapping or breathing of the nucleosome[38–40]. We examined the impact of the MORF$_{WH2}$ binding to the dyad on the nucleosome dynamics and unwrapping-wrapping equilibrium by Förster Resonance Energy Transfer (FRET)[41]. We prepared NCP$_{273}$ using 273 bp DNA, which contains the 601 sequence flanked by 50/76 bp linkers at the 5'/3' ends without the CpG-binding site for MORF$_{WH1}$ and the Cy3 donor fluorophore positioned 54 bp from the 5' end. The Cy5

**Fig. 5 | CpG recognition is conserved in MOZ/MORFWH1. a** Representative images of genomic localization of endogenous proteins and the indicated FLAG-tagged MOZ proteins. FLAG-tagged MOZ proteins were transiently expressed in HEK293T cells and analyzed by ChIP-seq. ChIP/CIRA-seq data at the *HOXA, MYC, CDKN2C,* and *CD4* genes are visualized using the Integrative Genomics Viewer (The Broad Institute). ChIP-seq analysis for MOZ_WH1 was performed in two independent replicates. The data range for each sample is shown on the right. **b** Average distribution of MOZ _WH1_ and MORF _WH1_ near the TSSs. FLAG-tagged MOZ _WH1_ and MORF _WH1_ were transiently expressed in HEK293T cells and

analyzed by ChIP-seq as in Fig. 4a. **c** Representative images of genomic localization of the FLAG-tagged MOZ _WH1_ and MORF _WH1_. ChIP-seq analysis for MORF _WH1_ was performed in two independent replicates. **d** Occupancy of the indicated wild type and mutated MORF _WH1_. ChIP-qPCR was performed for the *CD4, MYC* and *HOXA9* gene loci using qPCR probes designed for the pre-TSS (−1 to −0.5 kb of TSS), TSS (0 to +0.5 kb of TSS) and post-TSS regions (+1 to +1.5 kb of TSS) of each gene. ChIP signals are expressed as a percentage of input (mean ± SD of technical replicates; *n* = 3). Source data are provided as a Source Data file.

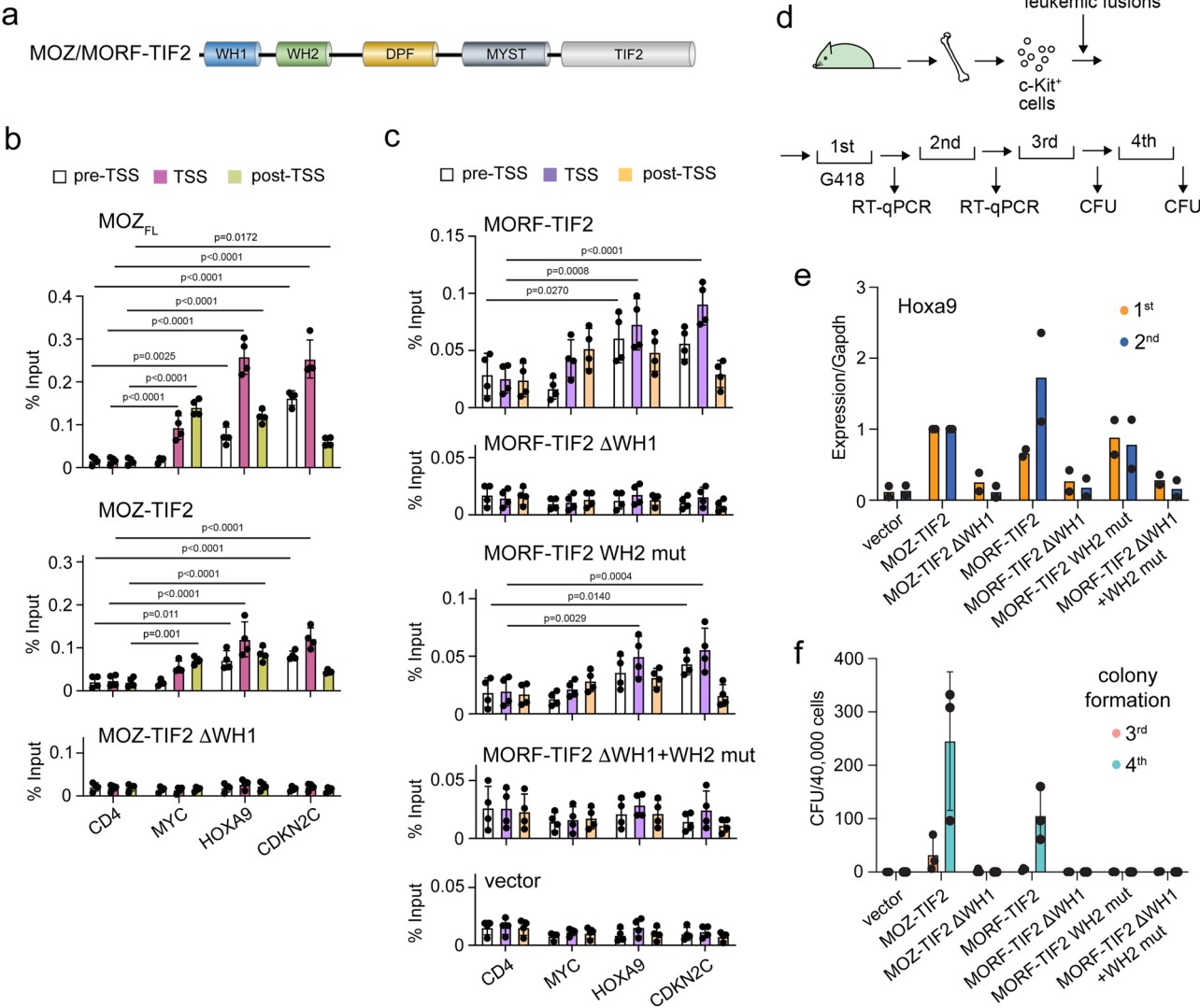

**Fig. 6 | MOZ/MORFWH1-mediated association with unmethylated CpG-rich gene promoters is essential for leukemic MOZ/MORF fusions. a** Schematic of the MOZ/MORF-TIF2 fusion. **b** Localization of the indicated MOZ-TIF2 proteins at *CD4, MYC, HOXA9,* and *CDKN2C* genes. ChIP-qPCR was performed for the indicated gene loci using qPCR probes designed for the pre-TSS (−1 to −0.5 kb of TSS), TSS (0 to +0.5 kb of TSS), and post-TSS regions (+1 to +1.5 kb of TSS) of each gene. ChIP signals are expressed as a percentage of input (mean ± SD of four technical replicates; *n* = 4). 2way ANOVA was performed and the statistical significance of the ChIP signals to the non-target gene CD4 is shown. *P* values are shown above bars. **c** Localization of the indicated MORF-TIF2 proteins at *CD4, MYC, HOXA9,* and

*CDKN2C* genes. ChIP-qPCR was performed and analyzed as in **b. d** A schematic of myeloid progenitor transformation assay. **e** Expression of *Hoxa9* in hematopoietic progenitors transfected with the indicated MOZ/MORF fusions. *Hoxa9* expression is normalized to *Gapdh* relative to MOZ-TIF2. Data represent mean of two biological replicates. Expression levels of MOZ-TIF2-transduced cells were arbitrarily set to 1 at each passage. **f** Leukemic transformation by MOZ/MORF fusions. Data represent mean ± SD of three biological replicates of Colony-forming units (CFUs) per 40,000 cells of hematopoietic progenitors transduced with the indicated MOZ/MORF fusions at the third and fourth passages. Source data are provided as a Source Data file.

acceptor fluorophore was attached to histone H2A(K119C) (Fig. 8f). Titration of His-MORF_WH1-WH2_ into NCP_273_ led to an increase in FRET efficiency, indicating stabilization of the wrapped state (Fig. 8g). We concluded that much like binding of linker H1/H5, binding of MORF_WH2_ to the nucleosome reduces DNA unwrapping. Altogether, cryo-EM,

FRET and EMSA results suggest a model for the cooperative association of two independent DNA-binding domains of MORF with the nucleosome, in which the typical MORF_WH2_ binds to the dyad of the nucleosome, whereas the atypical MORF_WH1_ associates with the CpG-containing linker DNA.

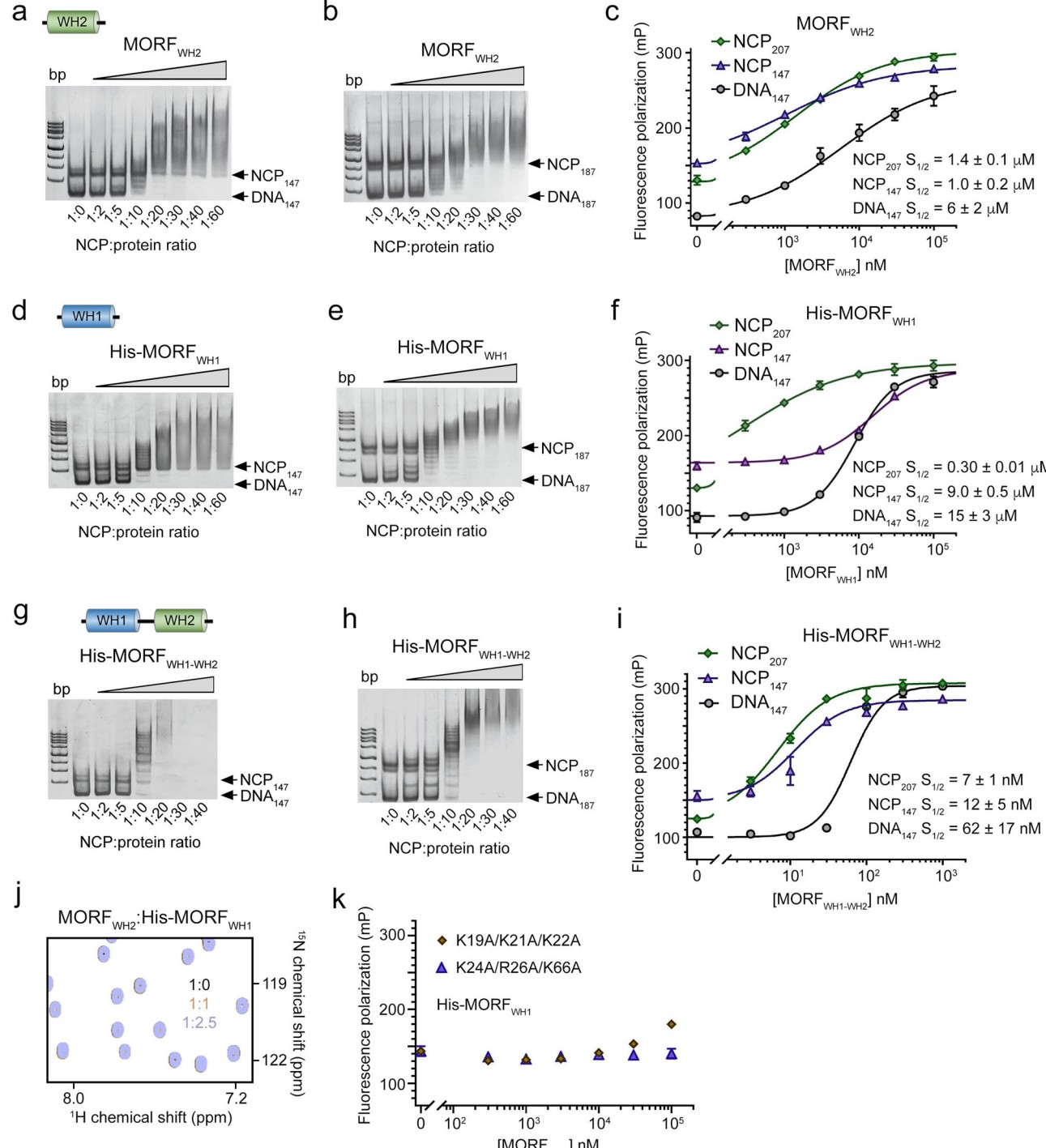

**Fig. 7 | Cooperative binding of MORFWH1-WH2 to the nucleosome. a, b** EMSAs of 147 bp NCP ($NCP_{147}$) and 187 bp NCP ($NCP_{187}$) in the presence of increasing amounts of $MORF_{WH2}$. NCP:protein ratio is shown below the gel images. **c** Binding curves for the interactions of $MORF_{WH2}$ with $NCP_{207}$, $NCP_{147}$, and $DNA_{147}$ as measured by fluorescence polarization. Data represent mean ± SD of three independent experiments. $n = 3$. **d, e** EMSAs of $NCP_{147}$ and $NCP_{187}$ in the presence of increasing amounts of His-$MORF_{WH1}$. NCP:protein ratio is shown below the gel images. **f** Binding curves for the interactions of $MORF_{WH1}$ with $NCP_{207}$, $NCP_{147}$ and $DNA_{147}$ as measured by fluorescence polarization. Data represent mean ± SD of three independent experiments. $n = 3$. **g, h** EMSAs of $NCP_{147}$ and $NCP_{187}$ in the presence of increasing amounts of MORF$_{WH1-WH2}$. NCP:protein ratio is shown below the gel images. **i** Binding curves for the interactions of MORF$_{WH1-WH2}$ with $NCP_{207}$, $NCP_{147}$, and $DNA_{147}$ as measured by fluorescence polarization. Data represent mean ± SD of three independent experiments. $n = 3$. **j** Overlay of $^1H,^{15}N$ HSQC spectra of $^{15}N$-labeled $MORF_{WH2}$ in the presence of increasing amount of unlabeled His-$MORF_{WH1}$. Spectra are color coded according to the protein:ligand molar ratio. **k** Fluorescence polarization of $NCP_{147}$ observed upon titration with the indicated $MORF_{WH1}$ mutants. Data represent mean ± SD of three independent experiments. $n = 3$ Source data are provided as a Source Data file.

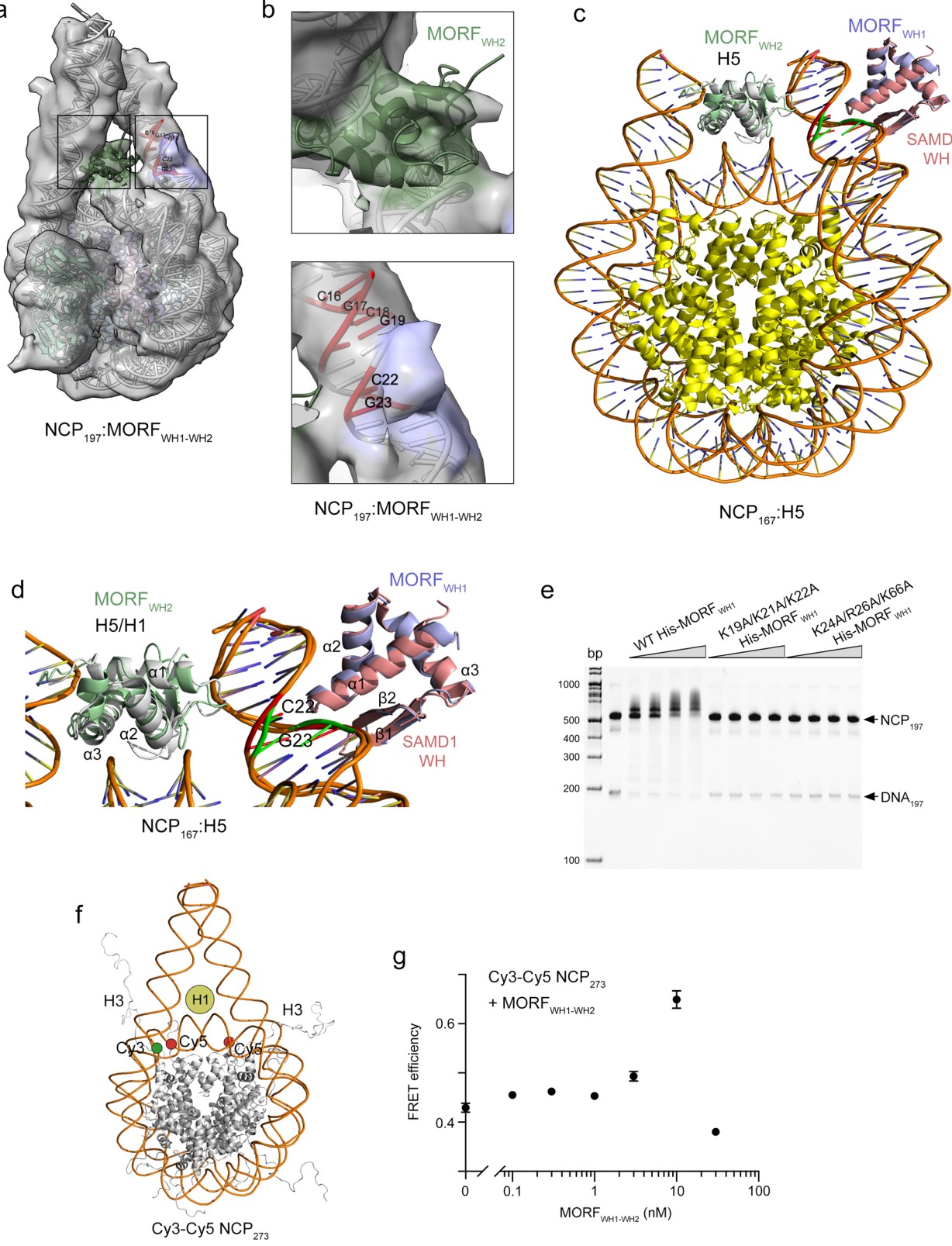

**MORF_WH1 binding to the linker DNA increases HAT activity**

The selectivity of MORF_WH1 toward the extra-nucleosomal linker DNA was observed in in vitro HAT assays. The native MORF complexes, containing 3xFLAG-2xStrep tagged MORF_N, WT or the deletion mutants of MORF_N (ΔWH1, ΔWH2, and ΔWH1/2), were affinity purified from nuclear extracts of K562 cells, and acetyltransferase function of these complexes was assessed on NCP_147 and NCP_207 (Fig. 9a and

Supplementary Fig. 11). The catalytic activity of the complex containing WT MORF_N was increased on the nucleosome with the linker DNA, NCP_207, compared to the activity of this complex on the nucleosome without the linker DNA, NCP_147. The deletion of MORF_WH1 or both WHs abolished the selectivity of the MORF_N complexes toward NCP_207, whereas the deletion of MORF_WH2 had little effect on the selectivity. These results imply that although both functional WHs are critical,

**Fig. 8 | A model for the association of MORFWH1-WH2 with the nucleosome.** **a**, **b** Cryo-EM map (transparent surface representation) of the MORF$_{WH1-WH2}$-197 bp nucleosome-scFv complex shows extra density near the nucleosome dyad region (map threshold level at 0.15). DNA and histones of 197 bp chromatosome structure (PDB ID: 7K5X) and MORF$_{WH2}$ (green) were docked into the density. The extra density observed on the linker DNA is colored blue. CpGs are colored red and labeled. **c** A model for the interaction of MORF$_{WH1-WH2}$ with the nucleosome. MORF$_{WH2}$ (green) is superimposed with H5 (gray) from the structure of NCP$_{167}$ in complex with H5 (PDB ID: 4QLC), and the AF model of MORF$_{WHI}$ from UniProt (Q8WYB5) (blue) is superimposed with the crystal structure of the DNA-bound WH from SAMD1 (salmon) (PDB ID 6LUI), as in Fig. 3i. The CpG sequence in the SAMD1-DNA structure is colored green, and the CpG sequence (C22, G23 in **a**, **b**) in NCP is colored red. **d** A zoom-in view of the model shown in **c** with the secondary structure elements labeled. **e** EMSA of 197 bp NCP (NCP$_{197}$) in the presence of increasing amounts of indicated WT or mutated His-MORF$_{WHI}$. **f** A model of Cy3-Cy5 labeled NCP$_{273}$ with Cy5 (red circles) positioned at H2AK119C and Cy3 (green circle) positioned at 54 bp from the DNA 5' end. Histone H1 bound at the dyad is depicted as tan circle. **g** FRET efficiency of Cy3-Cy5 NCP$_{273}$ upon addition of His-MORF$_{WH1-WH2}$. Data represent mean ± SD of at least three separate independent experiments. $n \geq 3$ A similar increase in FRET efficiency is observed due to binding of the linker histone H1[40]. Either H1[40] or His-MORF$_{WH1-WH2}$ above 10 nM induce nucleosome self-association, leading to a reduction in the FRET efficiency (Supplementary Fig. 10). Source data are provided as a Source Data file.

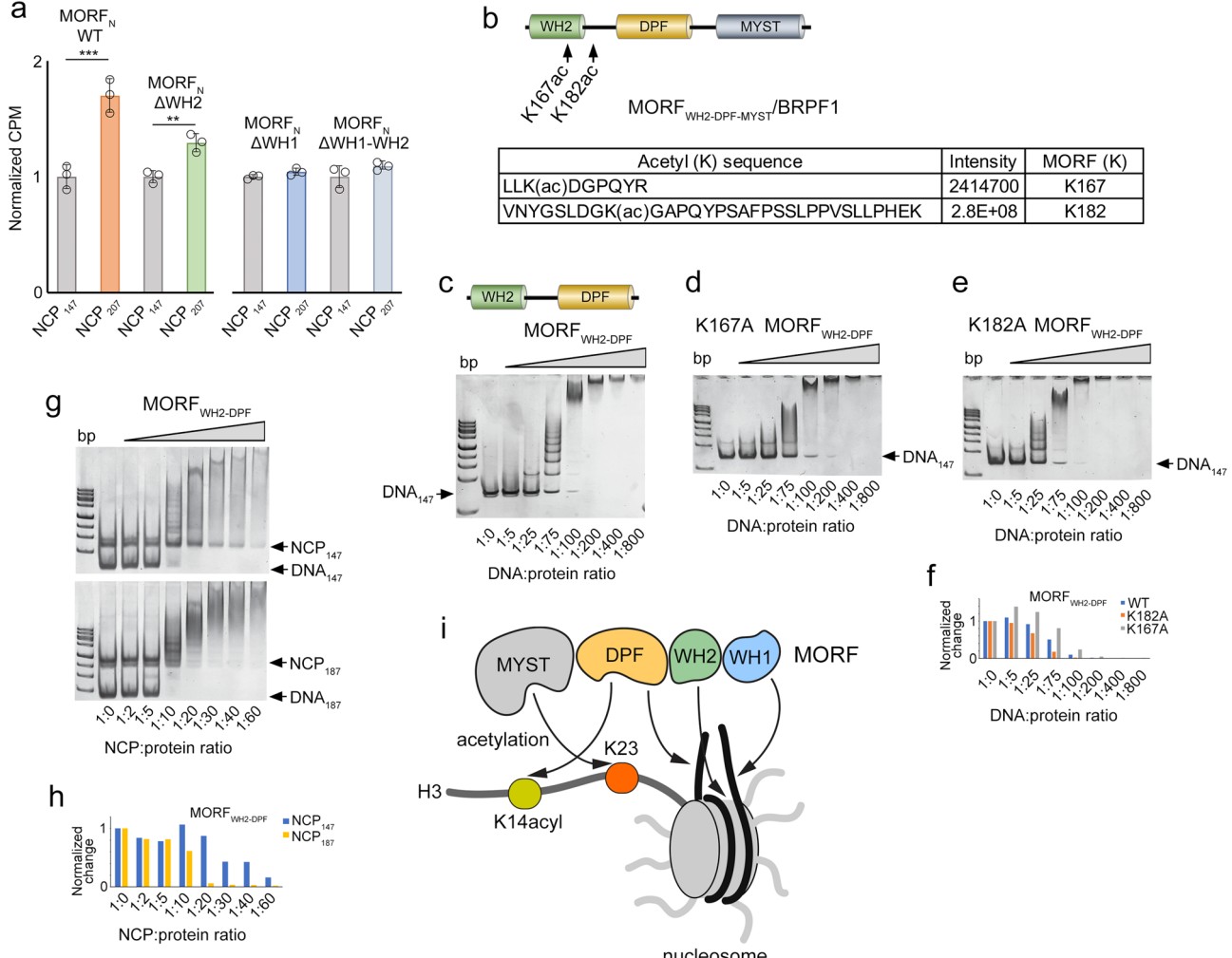

**Fig. 9 | Impact of auto-acetylation and the DNA linker on the HAT activity and binding to DNA. a** In vitro liquid HAT assays show acetyltransferase activity of native WT and mutated MORF$_N$ complexes, purified from K562 cells on NCP$_{207}$ and NCP$_{147}$. Values for NCP$_{207}$ are normalized to the corresponding values for NCP$_{147}$. Data represent mean ± SEM from three independent experiments. n = 3 Statistical tests were two-sided. Student's $t$-test, ***$P < 0.005$, $0.005 < **P < 0.01$ and $0.01 < *P < 0.05$. **b** Acetylated MORF$_{WH2-DPF-MYST}$ sequences detected by MS. **c–e** EMSAs of 147 bp DNA in the presence of increasing amounts of WT or mutated MORF$_{WH2-DPF}$. DNA:protein ratio is shown below the gel images. **f** Quantitative densitometry analysis of the DNA$_{147}$ band in **c–e**. **g** EMSAs of NCP$_{147}$ and NCP$_{187}$ in the presence of increasing amounts of MORF$_{WH2-DPF}$. NCP:protein ratio is shown below the gel images. **h** Quantitative densitometry analysis of the NCP$_{147}$ and NCP$_{187}$ bands in **g**. **i** A model for the interaction of MORF$_{WH1-WH2-DPF-MYST}$ with the nucleosome. Source data are provided as a Source Data file.

strong acetyltransferase activity of MORF depends on the interaction of MORF$_{WHI}$ with extra-nucleosomal DNA to a higher degree.

**MORF$_{WH2-DPF}$ binding to DNA is modulated by auto-acetylation**
MORF has been shown to be acetylated in the post-MYST region[42]. To test whether MORF self-acetylates its N-terminus, we co-purified catalytically active MORF$_{WH2-DPF-MYST}$ and BRPF1 and assessed acetylation by liquid chromatography-mass spectrometry (LC-MS). As control,

MORF$_{WH2-DPF-MYST}$/BRPF1 was treated with the NAD$^+$-dependent histone deacetylase SIRT2 to remove acetylation. We found two auto-acetylation sites in MORF$_{WH2-DPF-MYST}$, K167, located in MORF$_{WH2}$ and K182, located in the linker between MORF$_{WH2}$ and MORF$_{DPF}$ (Fig. 9b). We mutated both lysine residues to alanine and examined DNA binding activity of the K167A and K182A mutants of MORF$_{WH2-DPF}$ in EMSA (Fig. 9c–e). Quantitative densitometry analysis of the DNA$_{147}$ band revealed that while the K182A mutation increases binding of

MORF$_{WH2\text{-}DPF}$ to DNA$_{147}$, the K167A mutation decreases it (Fig. 9f). These data imply that autoacetylation of MORF can mediate its association with chromatin. EMSA assays further showed that MORF$_{WH2\text{-}DPF}$ prefers the nucleosome with an extra-nucleosomal linker DNA, as MORF$_{WH2\text{-}DPF}$ more readily forms the complex with NCP$_{187}$ than with NCP$_{147}$ (Fig. 9g, h). These results highlight the ability of MORF$_{WH1}$ and MORF$_{DPF}$, the two domains flanking MORF$_{WH2}$, to select for the linear, free of the nucleosome DNA.

In conclusion, in this study we identified a tandem of winged helix domains, WH1 and WH2, in the N-terminal regions of the human acetyltransferases MORF and MOZ. We found that both WHs interact with DNA but display selectivity for distinct sequences and use dissimilar mechanisms to engage DNA. The atypical WH1 binds exclusively to unmethylated CpG sequences through its α1 and two loops, whereas WH2 belongs to the family of typical WHs, has only a slight preference for the AT sequences, and associates with DNA via α3 and α2. In vivo data analyses reveal that the DNA binding of WHs (particularly of WH1) is essential for the recruitment of MORF/MOZ to promoters of target genes, stimulation of gene transcription and H3K23 acetylation, and thus is vital to physiological functions of these acetyltransferases. The WH1-mediated binding to unmethylated CpG-islands and the intact WH2 are also required for leukemogenic activity of the MOZ and MORF fusions. These results suggest a strategy directed at the inhibition of oncogenic MOZ and MORF translocations through targeting their WHs.

Two MORF$_{WHs}$ are followed by MORF$_{DPF}$ that was shown to bind H3K14acyl and DNA and the catalytic MORF$_{MYST}$ domain that acetylates H3K23[21,25]. The multivalent contacts of the four sequentially connected domains of MORF with DNA and histones and autoacetylation of MORF point to an intricate mechanism by which MORF targets specific sites and is activated there or inactivated when its catalytic activity is no longer needed. Our structural and biochemical findings suggest a model for the engagement of MORF with chromatin, in which MORF$_{WH1}$ binds to unmethylated CpG-containing linker DNA, MORF$_{WH2}$ occupies the dyad of the nucleosome, and MORF$_{DPF}$ interacts with H3K14acyl and the linker DNA (Fig. 9i). How the combination of these interactions stabilizes the MORF complex at chromatin and how autoacetylation, intermolecular contacts, and the presence of other subunits in the complex, such as ING4/5 and BRPF1 that recognize H3K4me3, H3K36me3, and acetylated histones[5], mediate function of the MORF/MOZ complexes, require further investigation.

## Methods

### Protein purification

The human MORF$_{WH1}$ (aa 5–84, His- and GST-tagged), MORF$_{WH2}$ (aa 100–182, His- and GST-tagged), MORF$_{WH1\text{-}WH2}$ (aa 1–182, His-tagged), MORF$_{WH2\text{-}DPF}$ (aa 100–322), MORF$_{DPF}$ (aa 211–322), BRPF1 (aa 89–139), MORF$_{WH2\text{-}DPF\text{-}MYST}$ (aa 100–703 with additional 3 lysine residues at the C-terminus), His-MOZ$_{WH1}$ (aa 1–86), GST-MOZ$_{WH1}$ (aa 2–86), MOZ$_{WH2}$ (aa 92–177, both GST- and His-tagged), and His-MOZ$_{WH1\text{-}WH2}$ (aa 1–177) constructs were cloned into pGEX-6P-1, pET22b, pET28a, pDESTsumo, pDONR, pDEST15 or pDEST17 vectors. The mutant constructs MORF$_{WH1}$ (K19A/K21A/K22A), MORF$_{WH1}$ (K24A/R26A/K66A), MORF$_{WH2}$ (K127A/K131A), MORF$_{WH2}$ (R151A, R153A, K157A, and R158A), MORF$_{WH2\text{-}DPF}$ (K167A), and MORF$_{WH2\text{-}DPF}$ (K182A) were generated using the Agilent QuikChange Lightning Site-Directed Mutagenesis kit. All constructs were confirmed by DNA sequencing. Unlabeled and $^{15}$N-labeled proteins were expressed in E. coli Rosetta-2 (DE3) pLysS cells grown in LB, TB, or $^{15}$NH$_4$Cl (Sigma-Aldrich) minimal media supplemented with ZnCl$_2$. After induction with IPTG (final concentration 0.2–0.5 mM, Gold biotechnology) for 16–20 h at 16 °C, cells were harvested via centrifugation and lysed in buffer (1× PBS or 25–50 mM Tris-HCl pH 7.0–7.5, 150–500 mM NaCl, 0.05% (v/v) Nonidet P 40, 0–10% glycerol, 5 mM dithiothreitol (DTT) or 2.5 mM (BME), phenylmethanesulfonyl-fluoride (PMSF), and DNase) by freeze-thaw followed by sonication.

The unlabeled and $^{15}$N-labeled GST-fusion proteins were purified on glutathione agarose beads (Thermo Fisher Sci). The GST-tag was cleaved with PreScission, tobacco etch virus (TEV) protease or Thrombin protease (MP Biomedicals) overnight or left on, and the proteins were eluted off the resin with 50 mM reduced L-glutathione (Fisher). His-tag fusion proteins were purified using nickel–NTA resin (ThermoFisher), the proteins were eluted from the resin with a gradient of imidazole. The His-tag was either cleaved with TEV protease during dialysis overnight at 4 °C or left on. When necessary, proteins were further purified by size exclusion chromatography (SEC) and concentrated in Millipore concentrators. His-SUMO-SIRT2 (aa 38–356) was purified as described[43]. For protein expression in HEK293T cells, cDNAs were obtained from Kazusa Genome Technologies Inc. or DNA fragments were synthesized and cloned into the pMSCV (for retrovirus production) or pCMV5 (for transient expression) vectors as described[13].

### DNA purification

Double-stranded DNA containing the 601 Widom sequence cloned into the pJ201 plasmid (147 bp) was transformed into DH5α cells. The plasmids were purified either as previously described[44] or by the PureLink HiPure Expi Plasmid Gigaprep Kit (Invitrogen K210009XP). Separation of the individual sequences was completed by digestion of the plasmid with EcoRV followed by PEG and ethanol precipitation. Short DNAs were either purchased as single-stranded DNA (IDT) and annealed or ordered as pre-annealed double-stranded DNA (IDT). Complimentary DNA strands were combined in a 1:1 molar ratio in water in PCR tubes. Using a thermocycler to regulate temperature, the samples were brought to 95 °C for 20 min and then to 16 °C at a rate of 0.1 °C/s. All DNA samples were evaluated for purity by native poly-acrylamide gel electrophoresis.

### EMSA

Increasing amounts of WT or mutant (tagged or untagged) MORF and MOZ proteins were incubated with DNA$_{147}$ (0.1–0.25 pmol/lane, 601 Widom sequence), DNA$_{15/16}$ (0.25–1.0 pmol/lane), DNA$_{ladder}$ (25 ng/lane) or NCP (0.5 pmol/lane) in buffer (20–25 mM Tris-HCl pH 7.5, 20–150 mM NaCl, 0–0.2 mM ethylenediaminetetraacetic acid (EDTA), 0–5 mM DTT, and 0–20% glycerol) in a 10 μL reaction volume. For the samples containing the DNA ladder (O'RangeRuler 5 bp DNA Ladder, ThermoSci) the buffer was 25 mM Tris-HCl pH 7.5, 150 mM NaCl. Reaction mixtures were incubated at 4 °C for 10 min (2.5 μl of loading dye was added to each sample) and loaded onto a 5–10% native poly-acrylamide gel. Electrophoresis was performed in 0.2 × TBE buffer (1 × TBE = 90 mM Tris, 64.6 mM boric acid, and 0–2 mM EDTA) at 80–130 V on ice. Gels were stained with SYBR Gold (Thermo Fisher) and visualized by Blue LED (UltraThin LED Illuminator- GelCompany). Uncropped gels are shown in the Source Data file.

Quantification of gel bands was performed using ImageJ using at least three independent experiments. $K_d$ values were determined using a nonlinear least-squares analysis and the equation:

$$\Delta I = \Delta I_{max} \frac{\left( ([P]+[D]+K_d) - \sqrt{([P]+[D]+K_d)^2 - 4[D][P]} \right)}{2[D]} \quad (1)$$

where $[P]$ is the concentration of the MORF protein, $[D]$ is the concentration of DNA, $\Delta I$ is the observed change of band intensity, and $\Delta I_{max}$ is the difference in band intensity of the free DNA and DNA bound by the protein. $K_d$ values were averaged over at least three separate experiments, and error was calculated as the standard deviation between the runs.

Binding of His-MORF$_{WH1}$, WT and mutants, to NCP$_{197}$ was monitored in buffer containing 10 mM Tris, 1 mM EDTA, 10 mM NaCl, 1 mM TCEP. 200 nM nucleosome was titrated with 0.4, 0.8, 1.2 and 2.0 μM

(replicate 1) or 1.6 µM (replicate 2) of His-MORF$_{WHI}$. Reactions were loaded on a 5% acrylamide gel and electrophoresis in 0.2 × TBE buffer at 120 V and at 4 °C.

## PBM experiments

Purified GST-MORF$_{WH1}$ (aa 5–84), GST-MORF$_{WH2}$ (aa 100–182), GST-MOZ$_{WH1}$ (aa 2–86), and GST-MOZ$_{WH2}$ (aa 92–177) were assayed at 300 nM final concentration in the PBM binding reaction on 8x60K GSE 'all 10-mer universal' oligonucleotide arrays (AMADID #030236; Agilent Technologies, Inc.), and proteins were detected using Alexa488 conjugated anti-GST antibody (Invitrogen A-11131) at a dilution of 1:40. Double-stranding of the arrays and PBM experiments were otherwise performed as described previously, and PBM data were quantified and analyzed using the PBM Universal Analysis Suite[31,32].

## Cell lines

Isogenic K562 cell lines expressing 3xFlag2xStrep-tagged MORF$_N$ WT (aa 2–716), ΔWH1 (aa 86–716), ΔWH2 (aa 2–99 + aa 183–716) and ΔWH1/2 (aa 183–716) were generated by integration at the AAVS1 safe harbor locus after DSB induction and recombination targeted by co-transfection with a ZFN expression plasmid, as previously described[45]. The forwards primers for WT For 5′- atatagcggccgcttc caccATGGTAAAACTTGCAAAC, ΔWH1 For 5′- atatagcggccgcttccaccA TGGGCACTTTTCCTAAGTCA and ΔWH1/2 For 5′- atatagcggccgcttcc accATGGGGGCACCTCAGTATCCC were used with the Rev 5′- atatagg ccggcCTCTTTCTCAGCTTCTCG. The MORF ΔWH2 (aa2–99 + 183–716) was generated by PCR amplification of the WT MORF (aa2–716) using For 5′- GGGGTCTAGAGGATCATGTGGGGCACCTC and Rev 5′- GAG GTGCCCCACATGATCCTCTAGACCCC primers designed by the Quick Change Primer Design-Agilent. $2 \times 10^5$ cells were transfected with 400 ng of ZFN expression vector and 4 µg of donor constructs. Selection and cloning were performed in RPMI medium supplemented with 0.5 µg/mL puromycin starting 2–3 days post transfection. Clones were obtained by limiting dilution and expanded before harvest for western blot analysis.

HEK293T cells were purchased from ATCC. Cells were cultured in Dulbecco's modified Eagles medium (DMEM), supplemented with 10% fetal bovine serum (FBS) and penicillin-streptomycin (PS). The platinum-E (PLAT-E) ecotropic virus-packaging cell line (a gift from Toshio Kitamura) was cultured in DMEM supplemented with 10% FBS, puromycin, blasticidin, and PS. Cells were cultured in an incubator at 37 °C and 5% $CO_2$ and routinely tested for mycoplasma using a MycoAlert Mycoplasma Detection Kit (Lonza).

## ChIP

Chromatin preparation from K562 cells was performed as previously described[46]. For chromatin immunoprecipitation, 150 µg of chromatin (for FLAG ChIP) and 50 µg of chromatin (for histones ChIP) was incubated with 3 µg anti-FLAG M2 (F1804, Sigma) or 1 µg anti-H3 (ab1791, Abcam), anti-H3K14ac (07–353, Upstate) and anti-H3K23ac (07–355, Upstate) antibodies overnight at 4 °C. 50 µl of Protein G Dynabeads for FLAG ChIP or 25 µl of Protein A Dynabeads were then added to each sample, and the mixtures were incubated at 4 °C for 4 h. The beads were washed extensively and eluted with 1% SDS and 0.1 M NaHCO3. Cross-linked samples were reversed by heating overnight at 65 °C in the presence of 0.2 M NaCl. Samples were then treated with RNase A and proteinase K for 2 h, and DNA was recovered using MinElute PCR purification Kit (Qiagen, 28004) according to the manufacturer's instructions. Quantitative real-time PCR corrected for primer efficiencies in the linear range was performed using SYBR Green I (Roche, 04877352001) on a LightCycler 480 (Roche). Expression levels of MORF$_{N1-716}$ WT, ΔWH1, ΔWH2 and ΔWH1/2 were monitored by running SDS-PAGE and transferring onto nitrocellulose membrane. Anti-FLAG M2 conjugated to horseradish peroxidase (A8592, Sigma) was used at 1:10,000 dilution.

Immunoblots were visualized using a Western Lightning plus ECL reagent (Perkin-Elmer).

## RT-qPCR

RNA was prepared using the RNeasy kit (Qiagen) and reverse transcribed using a Superscript III First Strand cDNA Synthesis kit, with oligo(dT) primers (Life Technologies). Gene expression was confirmed by qPCR using TaqMan probes (Life Technologies). Expression levels, normalized to those of *Gapdh*, were determined using a standard curve and the relative quantification method as described in ABI User Bulletin #2.

## Fractionation-assisted native chromatin immunoprecipitation (fanChIP)

Chromatin fractions from HEK293T cells were prepared using the fanChIP method as previously described[47]. Cells were suspended in CSK buffer and centrifuged to remove the soluble fraction in the same manner as the nucfrIP analysis. The pellet was resuspended in MNase buffer and treated with MNase at 37 °C for 3–6 min to obtain oligo-nucleosomes. The MNase reaction was stopped by adding EDTA (pH 8.0) to a final concentration of 20 mM. Lysis buffer (250 mM NaCl, 20 mM sodium phosphate [pH 7.0], 30 mM sodium pyrophosphate, 5 mM EDTA, 10 mM NaF, 0.1% NP-40, 10% glycerol, 1 mM DTT, and EDTA-free protease inhibitor cocktail) was added to increase solubility. The chromatin fraction was cleared by centrifugation and subjected to immunoprecipitation with specific antibodies [FLAG (Sigma-Aldrich F3165/M2, 1:400 dilution), MOZ (active motif 39868, 1:400 dilution), MEAF6 (STJ 116836, 1:400 dilution), ING4 (Abcam 108621, 1:400 dilution), Histone H3K14ac (Abcam ab52946, 1:400 dilution), RNAP2 non-P (Abcam 8WG16/ab817, 1:400 dilution), RNAP2 Ser5-P (Millipore CTD4H8/05-623, 1:400 dilution)] and magnetic microbeads (Protein-G magnet beads [Invitrogen]). Immunoprecipitates were washed five times with washing buffer (1:1 mixture of lysis buffer and MNase buffer with 20 mM EDTA) and then eluted in elution buffer. The eluted material was analyzed by qPCR and deep sequencing.

## ChIP-qPCR and ChIP-seq

The eluted material obtained by fanChIP was extracted by phenol/chloroform/isoamyl alcohol. DNA was precipitated with glycogen (Nacalai Tesque), dissolved in TE, and analyzed by qPCR and deep sequencing. For deep sequencing, Purified DNA was further fragmented (-150 bp long) using the Covaris M220 DNA shearing system (M&M Instruments Inc.). Deep sequencing was performed using a TruSeq ChIP Sample Prep Kit (illumina) and HiSeq2500 (illumina) at the core facility of Hiroshima University and the University of Tokyo. Data were visualized using the Integrative Genome Viewer (Broad Institute). Raw reads in fastq format were trimmed using cutadapt and aligned to the reference genome hg19 with BWA[48,49]. The alignment tags were counted, and ppm was calculated every 25 bp from TSS and TES of the genes. Heatmaps of ChIP signals on each TSS were generated by ngsplot[50]. Quantitative PCR (qPCR) analysis of the precipitated DNA was performed using the custom-made primer sets listed in Supplementary Tables 2 and 3. The values relative to inputs were determined using a standard curve and the relative quantification method.

## CpG island recovery assay

CpG island recovery assays for unmethylated CpGs (CIRA) were performed using the Unmethyl Collector kit (Active Motif) according to the manufacturer's instruction[13]. Briefly, genomic DNAs were prepared from HEK293T cells using DNeasy Blood & Tissue Kit (QIAGEN) and fragmented to the average size of 1 kb by sonication. The sonicated DNAs (100 ng) were incubated with the Histidine-tagged recombinant CXXC domain (6.5 µg) and magnetic nickel beads for 30 min at room temperature in the complete AM8 buffer supplied in the kit. The

reaction mixture was washed four times with the complete AM8 buffer and then eluted with the AM3 elution buffer supplied in the kit. For deep sequencing, purified DNA was further fragmented to the average size of 150 bp by sonication using a Covaris M220 DNA shearing system (M&M Instruments Inc.) and sequenced as described above.

## Virus production

Ecotropic retrovirus was produced using PLAT-E packaging cells[51]. The supernatant medium containing the virus was harvested 24–48 h following transfection and used for viral transduction.

## Myeloid progenitor transformation assay

The myeloid progenitor transformation assay was carried out as previously described[52]. Bone marrow cells were harvested from the femurs and tibiae of 5-week-old female C57BL/6 mice (purchased from CLEA Japan, Inc). c-Kit-positive cells were enriched using magnetic beads conjugated with an anti-c-Kit antibody (Miltenyi Biotec, 1:50 dilution), transduced with a recombinant retrovirus by spinoculation, and then plated in a methylcellulose medium (Iscove's Modified Dulbecco's Medium, 20% FBS, 1.6% methylcellulose, and 100 μM β-mercaptoethanol) containing murine stem cell factors, interleukin-3, and granulocyte-macrophage colony-stimulating factor (10 ng ml$^{-1}$ of each). G418 (1 mg ml$^{-1}$) was added to the first round of culture to select for transduced cells. *Hoxa9* expression was quantified by RT-qPCR after the first round of culture. Colony-forming units (CFUs) were quantified per $10^4$ plated cells after 4–6 days in culture. This protocol was approved by the National Cancer Center Institutional Animal Care and Use Committee of the National Cancer Center, Tsuruoka, Japan.

## Microscopy protein–protein interaction

GST-tagged MORF$_{WH2}$ (100 μM) was incubated with glutathione Sepharose 4B beads (Thermo Fisher Sci) at 4 °C for 0.5–1 h then washed with buffer (50 mM Tris-HCl pH 7.5, 150 mM NaCl, and 5 mM BME). Buffer was removed and the beads were resuspended in 1:1 washing buffer. To prepare for imaging, 10 μM fluorescein (FAM)-labeled 37 bp dsDNA (10–20 μM) was incubated with 10 μM of the suspended beads for 0.5–1 hour at room temperature. Confocal images were acquired on a Zeiss Observer.Z1 inverted microscope using a 488 nm laser for the excitation and emission of FAM. Images were processed using ImageJ.

## NMR experiments

Nuclear magnetic resonance (NMR) experiments were performed at 298 K on Bruker 600 MHz and Varian 900 MHz spectrometers. The $^1$H,$^{15}$N HSQC spectra of 0.1–0.2 mM uniformly $^{15}$N-labeled WT or mutant proteins were collected in the presence of increasing amount of unlabeled proteins or DNA (IDT). NMR data were processed and analyzed with NMRPipe and NMRDraw as previously described[53]. Normalized chemical shift changes were calculated using the equation

$$\Delta\delta = \sqrt{(\Delta\delta H)^2 + (\Delta\delta N/5)^2}, \qquad (2)$$

where $\Delta\delta$ is the change in chemical shift in parts per million (ppm).

## Structure determination for MORF$_{WH2}$

NMR samples for structure determination contained 1.3 mM $^{13}$C/$^{15}$N-labeled MORF$_{WH2}$ were prepared in 25 mM Tris-HCl (pH 6.8) buffer, supplemented with 150 mM NaCl and 8% D2O. Backbone and side chain chemical shift assignments for MORF$_{WH2}$ were obtained by collecting and processing a set of triple resonance experiments (HNCACB, CBCA(CO)NH, CC(CO)NH, HBHA(CO)NH, HNCA) with nonlinear sampling. 3D $^{15}$N- and $^{13}$C-edited NOESY-HSQC (mixing time of 100 ms) were collected to obtain distance restraints.

Calculation of the structure of MORF$_{WH2}$ (aa 100–182) was carried out using interproton NOE-derived distance restraints and

dihedral angle restraints. NMR spectra were processed and analyzed with NMRDraw and CcpNmr Suite[54]. The program DANGLE in CcpNmr Suite was used to predict dihedral angles ψ and φ restraints. Hydrogen bonds were derived from characteristic NOE patterns in combination with dihedral angles. The structures were calculated and refined with XPLOR-NIH[55]. 100 structures were calculated, and the ensemble of 15 conformers with the lowest total energy was selected to represent MORF$_{WH2}$. The quality of the structures was validated using the program PROCHECK-NMR. The percentage of residues in the most favored, additionally allowed, generously allowed and disallowed regions is 86.1, 12.5, 1.4, and 0.0, respectively. The structural statistics are listed in Supplementary Table 1.

## Nucleosome assembly

Human H2A, H2B, H3.2, and H4 histone proteins were expressed in *Escherichia coli* BL21 (DE3) pLysS cells, separated from inclusion bodies and purified using SEC and ion exchange chromatography. Histones were then combined in 7 M guanidine HCl, 20 mM Tris-HCl pH 7.5, and 10 mM dithiothreitol in appropriate molar ratios and refolded into octamer by slow dialysis into 2 M NaCl, 20 mM Tris-HCl pH 7.5, 1 mM ethylenediaminetetraacetic acid (EDTA) pH 8.0, and 2 mM β-mercaptoethanol. The octamer was purified from tetramer and dimer by SEC. Octamer was then mixed with DNA (147 bp or 207 bp 601 Widom sequence) in 5–10 mM Tris pH 8.0, 2 M NaCl and 0.5–1.0 mM EDTA, and NCPs were reconstituted by slow desalting dialysis into 5–10 mM Tris pH 8.0 and 0.5–1.0 mM EDTA. DNAs used in fluorescence polarization were 147 bp 601 Widom DNA fluorescein-labeled on the 5' end (for NCP$_{147}$) and 207 bp DNA (147 bp 601 DNA flanked with 30 bp linker DNA on either side and internally labeled with fluorescein 27 bp in from the 5' end) (for NCP$_{207}$). NCPs were separated from free DNA via sucrose gradient purification. When necessary, NCPs were purified by SEC and peak fractions were pooled. All NCPs were confirmed by SDS and native-PAGE. NCP$_{187}$ was purchased from Epicypher.

## Fluorescence polarization

Fluorescence polarization measurements were carried out by mixing increasing amounts of His-MORF$_{WH1}$ (aa 5–84), WT and mutants, MORF$_{WH2}$ (aa 100–182), or His-MORF$_{WH1-WH2}$ (aa 1–182) with 5 nM NCP$_{207}$ or NCP$_{147}$ in 75 mM NaCl, 25 mM Tris-HCl pH 7.5, 0.00625% Tween20, and 5 mM dithiothreitol in a 30 μL reaction volume. The samples were loaded into a Corning round-bottom polystyrene plate and allowed to incubate at 4 °C for 30 min. The polarization measurements were acquired with a Tecan infinite M1000Pro plate reader by exciting at 470 nm and measuring polarized emission at 519 nm with 5 nm excitation and emission bandwidths. The fluorescence polarization was calculated from the emission polarized parallel and perpendicular to the polarized excitation light as described previously[56]. The data were then fit to a non-cooperative binding isotherm to determine S$_{1/2}$. The S$_{1/2}$ values were averaged over three separate experiments with error calculated as the standard deviation between the runs.

## Förster resonance energy transfer

The Cy3-Cy5 labeled NCP$_{273}$ with Cy5 positioned at H2AK119C and Cy3 positioned at 54 bp from the DNA 5' end were prepared using a protocol described in ref. [40]. To perform FRET efficiency measurements, 0.5 nM NCP$_{273}$ were incubated with 0–300 nM His-MORF$_{WH1-WH2}$ in T130 buffer (10 mM Tris, 130 mM NaCl, 10% glycerol, and 0.0075% TWEEN20) for 20 min at room temperature. Fluorescence spectra were collected on a FluoroMax 4 fluorometer (Horiba) by exciting Cy3 at 510 nm and Cy5 at 610 nm and measuring emission from 530 to 750 nm. RatioA method was used to calculate FRET efficiency[40] from six separate experiments.

## Trypsin digestion of MORF

6 µl of 31 µM MORF$_{WH2\text{-}DPF\text{-}MYST}$ (aa 100–703 with additional 3 lysine residues at the C-terminus) co-expressed and purified with BRPF1 (aa 89–139) was treated with or without SIRT2 at 4 °C overnight in 30 mM Tris·HCl pH 7.5, 300 mM NaCl, 5% glycerol, 5 mM DTT, 0.5–1 mM NAD, 5 mM MgCl$_2$, and 50 µM ZnCl$_2$. Samples were denatured, reduced, and alkylated in 5% (w/v) sodium dodecyl sulfate (SDS), 10 mM tris (2-carboxyethyl) phosphine hydrochloride (TCEP·HCl), 40 mM 2-chloroacetamide, 50 mM Tris pH 8.5 and boiled at 95 °C for 10 min. Samples were prepared for mass spectrometry analyses using the SP3 method. Carboxylate-functionalized speedbeads (GE Life Sciences) were added to protein samples. Acetonitrile was added to 80% (v/v) to precipitate protein and bind it to the beads. The protein-bound beads were washed twice with 80% (v/v) ethanol and twice with 100% acetonitrile. Lys-C/Trypsin mix (Promega) was added for 1:50 protease to protein ratio in 50 mM Tris pH 8.5 and incubated rotating at 37 °C overnight. To clean up tryptic peptides, acetonitrile was added to 95% (v/v) to precipitate and bind peptides to the beads. One wash with 100% acetonitrile was performed and tryptic peptides were eluted twice with 1% (v/v) trifluoroacetic acid (TFA), 3% (v/v) acetonitrile in water. Eluate were dried using a speed-vac rotatory evaporator.

## Liquid chromatography and tandem mass spectrometry (LC-MS/MS) analysis

For acetylation sites analysis, the trypsinized peptides for the control (−) SIRT (n = 1) and the acetylated (+) SIRT (n = 1) were resuspended in 0.1% TFA, 3% acetonitrile in water, of which approximately 1 picomole of the peptides for each sample was directly injected onto a Waters M-class column (1.7 µm, 120 A, rpC18, 75 µm × 250 mm) and gradient eluted from 2% to 40% acetonitrile over 40 min at 0.3 µL/minute using a Thermo Ultimate 3000 UPLC (Thermo Scientific). Peptides were detected with a Thermo Q-Exactive HF-X mass spectrometer (Thermo Scientific) scanning MS1 spectra at 120,000 resolution from 380 to 1580 m/z with a 45 ms fill time and 3E6 AGC target. The top 12 most intense peaks were isolated with 1.4 m/z window with a 100 ms fill time and 1E6 AGC target and 27% HCD collision energy for MS2 spectra collected at 15,000 resolution. Dynamic exclusion was enabled for 5 seconds. MS data raw files were searched against the single Uniprot sequence for MORF (Uniprot accession number Q8WYB5-3) using Maxquant v.1.6.14.0 using Trypsin/P protease cleavage specificity allowing for two missed cleavages. Cysteine carbamidomethylation was searched as a fixed modification, while methionine oxidation, protein N-terminal and lysine side chain acetylation were treated as variable modifications. The mass tolerances for the database search were 4.5 ppm for the precursors and 20 ppm for the MS2 fragment ions, the minimum peptide length was seven residues with no additional applied score cutoffs. Peptide and protein level FDR were set at 0.01.

## Cryo-EM sample preparation, data collection, and processing

197 bp Widom 601 nucleosome and *PL2-6* nucleosome antibody fragment (scFv) were prepared according to previous publication[39]. 197 bp DNA sequence (center 147 bp Widom 601 sequence underlined, CG sequence in the linker DNA colored in red):

GGGCTGGACCCTATACGCGGCCGCCCTGGAGAATCCCGGTGCC GAGGCCGCTCAATTGGTCGTAGACAGCTCTAGCACCGCTTAAACGCA CGTACGCGCTGTCCCCCGCGTTTTAACCGCCAAGGGGATTACTCCCT AGTCTCCAGGCACGTGTCAGATATATACATCCTGTGCATGTATTGAAC AGCGACCACCCC. 10 µM MORF$_{WH1\text{-}WH2}$ was added step-wise to 0.2 µM 197 bp nucleosome to a final 6:1 molar ratio (MORF$_{WH1\text{-}WH2}$:nucleosome) in 500 µl binding buffer (10 mM HEPES, 0.1 mM EDTA, 10 mM NaCl, 5 mM 2-mercaptalethanol). After incubating the mixture at room temperature for 30 min, 100 µl of 3 µM scFv was added to the complex (threefold excess of scFv relative to the nucleosome). The MORF$_{WH1\text{-}WH2}$-nucleosome-scFv complex was then dialyzed to binding buffer overnight and concentrated to 1 ~ 2 µM using

a 30 kDa cut-off centrifugal filter unit (Millipore). 4 µl of prepared complex was applied to Lacey 300 mesh carbon grids (Ted Pella), glow discharged with a easiGlow discharger (Ted Pella) for 1 min at 15 mA. The grids were blotted using Whatman filter paper with 15 s waiting time, 2–3 s blotting time, and 20 blotting force at 4 °C and 100% humidity, then flash frozen in liquid ethane using a Vitrobot Mark IV (Thermo Fisher Scientific). A total of 2204 cryo-EM images were collected on a Talos Arctica microscope (Thermo Fisher Scientific) at 200 kV with a Gatan K3 Summit direct detection camera at the NCI-Frederick cryo-EM Facility. A magnification of 56 K was used, yielding pixel size of 0.91 Å/pixel. The movie frames were recorded at a dose rate of 18e⁻/px/s for 2.4 s exposure with 40 frames. SerialEM[57] was used for automatic data collection with defocus values set at a range from −0.8 to −1.8 µm.

Cryo-EM data were processed using cryoSPARC v3 software package[58]. Movie frames were aligned with Patch motion correction and contrast-transfer function (CTF) estimation was performed by Patch CTF estimation tools with cryoSPARC live during the data collection. Particles picking were performed with Blob picker followed by Template picker tools. Initial picked 219800 particles were cleaned by one round of 2D classification, 66885 particles were selected for ab initio model generation. Non-Uniform refinement were then performed with the ab initio model and selected particles as input. A generous mask was generated using the non-uniform refined map. 3D variability analysis was performed using the generated mask and particles from non-uniform refinement. A cluster reconstructed from 4629 particles were selected with the 3D variability analysis (4 cluster mode) which shows extra density on the nucleosome dyad region. A final non-uniform refinement was performed with particles from the selected cluster, yielding an overall 7 Å cryo-EM map (See Supplementary Fig. 8). Cryo-EM maps were illustrated using UCSF ChimeraX (https://www.rbvi.ucsf.edu/chimerax).

## Purification of native MORF complexes

For purification of native complexes, after large-scale expansion of K562 clones, affinity purifications of tagged MORF$_N$ WT (aa 2–716), ΔWH1 (aa 86–716), ΔWH2 (aa 2–99 + aa 183–716) and ΔWH1/2 (aa 183–716), were performed on nuclear extracts as previously described[59]. Briefly, nuclear extracts were prepared following standard procedures and pre-cleared with CL6B Sepharose beads. FLAG immunoprecipitations with anti-FLAG agarose affinity gel (Sigma M2, 250 µl) were performed, followed by elution with 3xFLAG peptide (200 µg/mL from Sigma) in the following buffer: 20 mM HEPES pH 7.5, 150 mM KCl, 0.1 mM EDTA, 10% glycerol, 0.1% Tween20, 1 mM DTT and supplemented with proteases, deacetylases, and phosphatase inhibitors. Expression was measured by WB (Supplementary Fig. 3a) using anti-FLAG M2 (Sigma, F1804, 1:10,000 dilution) and anti-WDR5 (a gift from Edwin Smith, 1:1000 dilution) antibodies.

## In vitro HAT assays of the MORF$_N$ complexes

Acetyltransferase activity of the purified complexes containing MORF$_N$ WT (aa 2–716), ΔWH1 (aa 86–716), ΔWH2 (aa 2–99 + aa 183–716) or ΔWH1/2 (aa 183–716) was measured using 0.125 µCi of ³H labeled Ac-CoA (2.1 Ci/mmol; PerkinElmer Life Sciences) or 150 µM of unlabeled (cold) Ac-CoA. The HAT reactions were performed in a volume of 15 µl using 0.5 µg of NCP$_{207}$ (30 bp linker DNA flanking both sides of 147 bp 601 Widom DNA) (produced as previously described[60]) and NCP$_{147}$, in HAT buffer (50 mM Tris·HCl pH 8, 50 mM KCl, 10 mM sodium butyrate, 5% glycerol, 0.1 mM EDTA, 1 mM dithiothreitol) for 30 min at 30 °C. The reactions were then captured on P81 filter paper, the free ³H-labeled Ac-CoA was washed away, and the paper was analyzed by liquid scintillation. For in vitro HAT assays with cold Ac-CoA, the HAT activity of WT and mutated MORF$_N$ complexes on specific sites of histone H3 were monitored by western blot. Wild type and mutant complexes were normalized by western blot and the HAT

activity on free histones. The following antibodies were used: anti-H3K23ac (Upstate, 07–355, 1:1000 dilution) and anti-H3 (Abcam, ab1791, 1:20,000 dilution).

## Statistics and reproducibility

Statistical analysis shown in Fig. 6 was performed using GraphPad Prism 8 and Microsoft Excel software. Data are presented as mean ± standard deviation (SD). EMSA experiments were performed at least twice. Multiple comparisons were performed using one-way or two-way ANOVA; all statistical tests were two-sided. Statistical significance was set at $P \leq 0.05$. n.s.: $P > 0.05$, *$P \leq 0.05$, **$P \leq 0.01$, ***$P \leq 0.001$, and ****$P \leq 0.0001$ (Fig. 1f–h), and ***$P < 0.005$, $0.005 < **P < 0.01$ and $0.01 < *P < 0.05$ by student's $t$-test (Fig. 9a).

## Reporting summary

Further information on research design is available in the Nature Portfolio Reporting Summary linked to this article.

## Data availability

The data that support this study are available from the corresponding authors upon reasonable request. Coordinates and structure factors have been deposited in the Protein Data Bank under accession code 8E4V. NMR data have been deposited in the Biological Magnetic Resonance Bank under accession number 31040. Cryo-EM map of the MORF$_{WH1-WH2}$ and 197 bp nucleosome complex has been deposited in the Electron Microscopy Data Bank under accession number EMD-27243. ChIP-seq and CIRA-seq data have been deposited to the DDBJ (DNA Data Bank of Japan) Sequence Read Archive as fastq files and as WIG files under accession numbers DRA008734, DRA012473, DRA008732, DRA014291, DRA014290, DRA010562, DRA015383, E-GEAD-324, E-GEAD-446, E-GEAD-322, E-GEAD-497, E-GEAD-498, E-GEAD-381 and E-GEAD-584 [https://ddbj.nig.ac.jp/public/ddbj_database/dra/fastq/] and [https://ddbj.nig.ac.jp/public/ddbj_database/gea/experiment/E-GEAD-000/] (Supplementary Table 4). The mass spec data have been deposited to the PRIDE database under accession number PXD036192. Source data are provided with this paper.

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

## Acknowledgements

We thank Tina Holt, Hagumu Sato, Ikuko Yokoyama, Kanae Ito, Etsuko Kanai, and Ayako Yokoyama for technical assistance, members of the Shonai Regional Industry Promotion Center for their administrative support, Stephen Gisselbrecht for discussion, and Dan Shi of the NCI-Frederick cryo-EM Facility for help with cryo-EM data collection. This work utilized the computational resources of the NIH HPC Biowulf cluster [http://hpc.nih.gov]. This work was supported in part by grants from the NIH: HL151334, GM135671, GM125195, CA252707, and AG067664 to T.G.K., GM131626 and GM139564 to M.G.P., and HG010501 to M.L.B., from the Japan Society for the Promotion of Science (JSPS) KAKENHI grants (19H03694, 22H03109 and 22KK0119) to A.Y., from the Canadian Institutes of Health Research (CIHR) (FDN-143314) to J.C. and from a Natural Sciences and Engineering Research Council of Canada (RGPIN-2016-05844) to A.F.T. M.G. is supported by a doctoral fellowship from the Fonds de Recherche du Québec - Santé (FRQS). This work was also supported in part by research funds from the Yamagata prefectural government and the city of Tsuruoka. B-R.Z. and Y.B. are supported by the intramural research program of the National Cancer Institute, NIH. The content is solely the responsibility of the authors and does not necessarily represent the official views of the NIH.

## Author contributions

D.C.B., B.J.K., A.K., S.M.J., K.L.C., B-R.Z., S.K.P., Y.Z., R-W.C., C.C.E., C.L., and M.G. performed experiments and together with A.F.T., M.L.B., Y.B., M.G.P., J.C., A.Y., and T.G.K. analyzed the data. A.Y. and T.G.K. wrote the manuscript with input from all authors.

## Competing interests

The authors declare no competing interests.
