## [Peer Review File · Nature Communications]

REVIEWER COMMENTS

Reviewer #1 (Remarks to the Author):

In the manuscript, Becht et al described their characterization of the DNA binding activities of the winged helix domains WH1 and WH2 of human acetyltransferases MORF and MOZ. Using the combined structural, biochemical and genomic sequencing analyses, the authors showed that the WH1 prefers to bind unmethylated CpG sequences, while WH2's DNA binding is less dependent on the DNA sequence but rather influenced by the DPF domain following the WH2 in MORZ/MOZ that is known to recognize histone H3K14ac. Furthermore, the authors demonstrated that the WH1 binds to CpG-rich linker DNA in the nucleosome context while the WH2 occupancies the dyad of the nucleosome, and the DPF interacts with WH1/2, histone H3K14ac, and the linker DNA. Collectively, this study provides new detailed understanding of how MORF/MOZ through their respective structural domains' interactions with nucleosomal DNA and lysine-acetylated histone H3 are recruited to target gene loci in chromatin and facilitate histone lysine acetylation associated with gene transcriptional expression. There are several issues, as described below, that the authors should address before the manuscript is suitable for publication.

Specific Comments:

1. The authors showed that MORF-WH1 prefers CpG sequence while MORF-WH2 does not have distinctive DNA sequence preference. However, the structural and molecular basis of DNA binding specificity between the WH1 and WH2 domains is lacking. It would be helpful that the authors can provide a model structure of WH1, which may be generated either from molecular modeling based on the structure of the WH2 domain or from the cryo-EM map. Such a model structure would also help explain how methylation of CpG would reduce the DNA binding of MORF-WH1.
2. As suggested by the data presented in Figure 7, the DPF domain possibly binds to lysine-acetylation sites in WH2 through exerting an auto-inhibition effect on WH2/DNA binding activity. The authors should assess whether DPF binding to H3K14ac would release this auto-inhibition effect on WH2/DNA binding in chromatin.
3. Quantification of the EMSA binding data for MORF-WH2-DPF wt vs. K182A or K167A mutant binding to DNA (Figure 7i) will be helpful to illustrate the auto-inhibition of DPF on WH2/DNA binding.
4. More data and discussion on the coordination between WH2/DNA binding and DPF/histone binding would further enhance this study.

Reviewer #2 (Remarks to the Author):

Histone acetylation plays an important role in gene regulation and is involved in the formation of relaxed chromatin, euchromatin. This structure enables transcription factors to access to the specific DNA sequence. MOZ/MORF acetyltransferase complexes play important roles in diverse biological events and are implicated in chromosomal translocations associated with leukemias.

In this paper, Becht et al identified the two DNA binding domains in the N-terminal part of MOZ/MORF. The structure of the DNA binding domains belongs to winged helix domain, named as WH1 and WH2. In vitro gel shift assay showed that the WH1 and WH2 of MORF bind to DNA duplex (synthesized nucleosome free DNA) and important for the MORF target

gene acetylation. The authors determined the solution structures of apo-WH1 and -WH2 of MORF2 by NMR and binding surface for DNA by titration experiments. To gain the sequence specificity of WH1 and WH2 of MORF1, the authors performed DNA-binding selectivity assay using universal oligonucleotide arrays, revealing that WH1 of MORF1 showed preference for unmethylated CpG sequence and WH2 of MORF1 had a slight preference for an AT-rich sequence, which were further confirmed by biochemical and fluorescence polarization assay. Next, the authors demonstrated that ChIP-Seq, CIRA-Seq and ChIP-qPCR showed that WH1 is important for the recruitment of MOZ and MOZ-oncogenic fusion protein to DNA promoters of target genes with enhancing transcription and H3K23 acetylation. Finally, the authors further performed the biochemical and structural studies (cryo-EM and NMR) of MORF using nucleosomes, which revealed the binding mode of WH1 and WH2 of MORF to nucleosome.

Understanding the molecular functions of MORF/MOZ is a very important for revealing the mechanism of transcriptional activation of target genes and understanding the pathogenesis of leukemias. However, it is not of sufficient quality for the publication because the manuscript contains unconvincing data, especially Fig 6 and 7. Please consider below for major and minor points.

[Major concerns]

1. Overall, the biochemical and structural studies were preformed using MORF protein. However, ChIP-Seq, CIRA-Seq and ChIP-qPCR were conducted using MOZ protein. Fig 6i showed the cooperative (simultaneous) binding of WH1 and WH2 of MORF to nucleosome and the cryo-EM indicates the cooperative binding of WH1 and WH2 of MORF to nucleosome. However, functional assay using ChIP-Seq, CIRA-Seq and ChIP-qPCR indicate that only WH1 of MOZ is sufficient for the target gene localization of MOZ. This discrepancy should be addressed in the revised manuscript.
In addition, Fig 1e shows that the linker length between WH1 and WH2 of MORF is different from that of MOZ. If this difference is the critical reason for the discrepancy, the entire experiments should be done with MOZ of MORF only. The authors should elaborate on the validity of not performing the entire experiments using the same protein. this issue.
2. WH1 of MORF has a preference for unmethylated CpG sequence DNA. However, it is unclear the molecular mechanism by which the WH1 can specifically bind to unmethylated CG sequence DNA. WH1 is much important for the biological function of MOZ than WH2; WH1 is responsible for localization at CG promoter of MOZ. Nevertheless, the manuscript did not address the molecular mechanism for the recognition of unmethylated CpG sequence DNA by WH1 domain of MOZ. The authors should determine the structure of WH1 in complex with unmethylated CG sequence DNA or should mention the possible molecular mechanism underlying the recognition of unmethylated CG DNA by WH1 of MORF and MOZ.
3. In Fig 6j, the authors concluded that there is not physical interaction between WH1 and WH2 of MORF. However, only a magnified view of a portion of the NMR spectrum is shown in the figure. The entire NMR spectrum should be presented to clearly show no spectrum changes during the titration experiment.
4. Cryo-EM data is not completely convincing for determining the spatial positions of WH1 and WH2 of MORF. The shape (size) of WH1 is similar to that of WH2. However, the authors modeled WH1 on the dyad and WH2 on the linker DNA. The resolution of the final model is low. Therefore, the densities corresponding to the WH1 and WH2 are obscure, and it is not possible to distinguish which densities are WH1 and WH2. I could not understand the rationale for the modeling of WH1 and WH2. In addition, information of linker DNA sequence is lacking. Does the linker DNA contain CG-rich sequence? If so, the design of the linker DNA is the reason of the modeling of WH1 and WH2? Anyway, the authors should perform cryo-EM experiment using the electron microscope with accelerating voltage of 300

kv and reconsider the experimental design, e.g., preparation of sample grid and cross-linking of the complex, etc., to obtain more clear cryo-EM map for the WH1 and WH2.

5. In Fig 7a, b and Supplementary Fig 6, the authors described that dissimilar pattern of CSPs of titration experiments. However, the pattern of the spectrum changes of G279, G310, S208, S290 are similar. The authors also claimed slow to intermediate exchange regime on the NMR time scale seen upon titration of MORF WH2 revealed that binding of MORF WH2 is stronger than binding of MORF WH1. However, judging from the Supplementary Fig 6, this spectrum showed typical fast to intermediate regime. In addition, the measurements were not conducted at the same concentration ratio, so it should not be compared. The authors should re-perform the NMR titration experiment at the same concentration ratio and should estimate the K_d values from the signal changes. Furthermore, identification of the binding surface of WH1 and WH2 for DPF-binding should be required for the arguing the different binding mode between WH1/WH2 and DPF. To do so, the authors should perform the inverse titration experiments using labeled WH1/WH2 and unlabeled DPF.

6, In Fig 7a and 7b, histogram of normalized CSPs, as shown in the Figure 2C, should be prepared in the revised manuscript.

7, In Fig 7h and 7i.

Fig 7h showed that DNA binding ability of WH2-DPF of MORF was reduced by DPF. Fig 7i showed that either K182A and K167A in WH2 led to increase in binding of WH2-DPF of MORF to DNA. However, Fig 7h was performed using GST-fused WH2-DPF of MORF. It can not be ruled out that the dimer formation of GST reduces the DNA binding ability of WH2-DPF of MORF. To compare under the same conditions, Fig 7h should be performed using untagged WH2-DPF of MORF.

8, The model structure in Fig 7m was not prepared based on experimental evidence. Simultaneous binding of WH1 and WH2 to DPF should be shown at least.

[Minor comments]

Cryo-EM

scFV was used for preparation of the Cryo-EM sample. It binds to the disk of the nucleosome, however, the possibility that binding of scFV inhibit the interaction between nucleosome and WH1-WH2 of MORF can not be excluded. EMSA should be performed for detecting the interaction between nucleosome and WH1-WH2 of MORF in the presence of scFV.

Reviewer #3 (Remarks to the Author):

This reviewer only focused on the DNA methylation analyses and epigenomic sequencing aspects of the manuscript.

The DNA methylation and histone modification sequencing of the manuscript seem ok, although I could find out how exactly the authors performed CpG island recovery assays for unmethylated CpGs (CIRA). They cited one of their earlier paper, but that paper also did not provide detailed protocol. Both referred to the Unmethyl Collector kit (Active Motif) for the CIRA assay, but I could not find any information on this kit and according to the authors' previous paper, the product was discontinued. So how can the authors still perform this assay or could they still perform this assay in the future? I think it would be good for the

authors to provide details protocol and cite the original paper for the CIRA assay, so other researchers could reproduce their results if needed.

We thank the Editor and Reviewers for the insightful and very constructive comments, which were helpful in revising and strengthening this manuscript.

Reviewer 1: In the manuscript, Becht et al described their characterization of the DNA binding activities of the winged helix domains WH1 and WH2 of human acetyltransferases MORF and MOZ. Using the combined structural, biochemical and genomic sequencing analyses, the authors showed that the WH1 prefers to bind unmethylated CpG sequences, while WH2's DNA binding is less dependent on the DNA sequence but rather influenced by the DPF domain following the WH2 in MORZ/MOZ that is known to recognize histone H3K14ac. Furthermore, the authors demonstrated that the WH1 binds to CpG-rich linker DNA in the nucleosome context while the WH2 occupancies the dyad of the nucleosome, and the DPF interacts with WH1/2, histone H3K14ac, and the linker DNA. Collectively, this study provides new detailed understanding of how MORF/MOZ through their respective structural domains' interactions with nucleosomal DNA and lysine-acetylated histone H3 are recruited to target gene loci in chromatin and facilitate histone lysine acetylation associated with gene transcriptional expression. There are several issues, as described below, that the authors should address before the manuscript is suitable for publication.

Specific Comments:

Reviewer 1, Comment 1: 1. The authors showed that MORF-WH1 prefers CpG sequence while MORF-WH2 does not have distinctive DNA sequence preference. However, the structural and molecular basis of DNA binding specificity between the WH1 and WH2 domains is lacking. It would be helpful that the authors can provide a model structure of WH1, which may be generated either from molecular modeling based on the structure of the WH2 domain or from the cryo-EM map. Such a model structure would also help explain how methylation of CpG would reduce the DNA binding of MORF-WH1.

Author's response: the alpha-fold predicted structure of WH1 is shown in Fig. 3h, and we used it to map the DNA binding interphase, however it will be risky to discuss the details of how methylation of CpG reduces binding using the modeling. We have set up numerous co-crystallization trays though haven't obtained diffracting crystals of the WH1-DNA complex.

Reviewer 1, Comment 2: 2. As suggested by the data presented in Figure 7, the DPF domain possibly binds to lysine-acetylation sites in WH2 through exerting an auto-inhibition effect on WH2/DNA binding activity. The authors should assess whether DPF binding to H3K14ac would release this auto-inhibition effect on WH2/DNA binding in chromatin.

Author's response: we continue studying the relationship between the WH1, WH2, DPF and MYST domains. The autoregulatory mechanism appears to be more complex than we initially thought and requires more vigorous characterization. To avoid premature conclusions and keep the manuscript focused on the major finding of this work, the discovery of two WHs and their DNA binding functions, we have removed these data from Fig. 7.

Reviewer 1, Comments 3 and 4: 3. Quantification of the EMSA binding data for MORF-WH2-DPF wt vs. K182A or K167A mutant binding to DNA (Figure 7i) will be helpful to illustrate the auto-inhibition of DPF on WH2/DNA binding.

4. More data and discussion on the coordination between WH2/DNA binding and DPF/histone binding would further enhance this study.

As suggested, we have quantified EMSAs, added in vitro HAT data with native complexes (new Fig. 7a) and fluorescence anisotropy data on MOZ (new Suppl. Fig. 5) and expanded the discussion on the interplay between histone and DNA binding activities of WHs and DPF.

Reviewer 2: Histone acetylation plays an important role in gene regulation and is involved in the formation of relaxed chromatin, euchromatin. This structure enables transcription factors to access to the specific DNA sequence. MOZ/MORF acetyltransferase complexes play important roles in diverse biological events and are implicated in chromosomal translocations associated with leukemias. In this paper, Becht et al identified the two DNA binding domains in the N-terminal part of MOZ/MORF. The structure of the DNA binding domains belongs to winged helix domain, named as WH1 and WH2. In vitro gel shift assay showed that the WH1 and WH2 of MORF bind to DNA duplex (synthesized nucleosome free DNA) and important for the MORF target gene acetylation. The authors determined the solution structures of apo-WH1 and -WH2 of MORF2 by NMR and binding surface for DNA by titration experiments. To gain the sequence specificity of WH1 and WH2 of MORF1, the authors performed DNA-binding selectivity assay using universal oligonucleotide arrays, revealing that WH1 of MORF1 showed preference for unmethylated CpG sequence and WH2 of MORF1 had a slight preference for an AT-rich sequence, which were further confirmed by biochemical and fluorescence polarization assay. Next, the authors demonstrated that ChIP-Seq, CIRA-Seq and ChIP-qPCR showed that WH1 is important for the recruitment of MOZ and MOZ-oncogenic fusion protein to DNA promoters of target genes with enhancing transcription and H3K23 acetylation. Finally, the authors further performed the biochemical and structural studies (cryo-EM and NMR) of MORF using nucleosomes, which revealed the binding mode of WH1 and WH2 of MORF to nucleosome. Understanding the molecular functions of MORF/MOZ is a very important for revealing the mechanism of transcriptional activation of target genes and understanding the pathogenesis of leukemias. However, it is not of sufficient quality for the publication because the manuscript contains unconvincing data, especially Fig 6 and 7. Please consider below for major and minor points.

[Major concerns]

Reviewer 2, Comment 1: 1. Overall, the biochemical and structural studies were performed using MORF protein. However, ChIP-Seq, CIRA-Seq and ChIP-qPCR were conducted using MOZ protein. Fig 6i showed the cooperative (simultaneous) binding of WH1 and WH2 of MORF to nucleosome and the cryo-EM indicates the cooperative binding of WH1 and WH2 of MORF to nucleosome. However, functional assay using ChIP-Seq, CIRA-Seq and ChIP-qPCR indicate that only WH1 of MOZ is sufficient for the target gene localization of MOZ. In addition, Fig 1e shows that the linker length between WH1 and WH2 of MORF is different from that of MOZ. If this difference is the critical reason for the discrepancy, the entire experiments should be done with MOZ of MORF only. The authors should elaborate on the validity of not performing the entire experiments using the same protein.

Author's response: there is no discrepancy- ChIP-seq experiments show that WH1 is necessary for targeting of MOZ to *unmethylated CpG*. The delta WH1 MOZ still binds chromatin though not specifically to the unmethylated CpG islands.

We note that we carried out functional and biochemical experiments with both MORF and MOZ. ChIP-qPCR (Figs. 5c, d and Fig. 1f) and transformation assays (Figs. 5f, g), EMSAs (Figs. 1c, d and Suppl. Fig. 2b), NMR (Fig. 1b and Suppl. Fig. 2a), PBM arrays (Figs. 3a, e and Suppl. Fig. 2c) show the same results and point to a high conservation of functions of these MOZ/MORF domains. It's not surprising, these homologous proteins are often referred to as MOZ/MORF because of their high functional and structural similarity.

Additionally, we have performed fluorescence polarization experiments using WH1-WH2 of MOZ. The new data (Suppl. Fig. 5) confirm that the MOZ WH1-WH2 behaves in a manner similar to that of MORF WH1-WH2, and the length of the linker between WH1 and WH2 (32 aa in MORF and 24 aa in MOZ) has no effect (Fig. 6i). The overlay of WH1-WH2 of MORF and MOZ generated using the alpha fold program (see Fig. below) further corroborates these results.

Reviewer 2, Comment 2: 2. WH1 of MORF has a preference for unmethylated CpG sequence DNA. However, it is unclear the molecular mechanism by which the WH1 can specifically bind to unmethylated CG sequence DNA. WH1 is much more important for the biological function of MOZ than WH2; WH1 is responsible for localization at CG promoter of MOZ. Nevertheless, the manuscript did not address the molecular mechanism for the recognition of unmethylated CpG sequence DNA by WH1 domain of MOZ. The authors should determine the structure of WH1 in complex with unmethylated CG sequence DNA or should mention the possible molecular mechanism underlying the recognition of unmethylated CG DNA by WH1 of MORF and MOZ.

Author's response: despite setting up numerous co-crystallization trays in the past three years we have failed to obtain diffracting crystals of the WH1-DNA complex; this work is in progress.

Reviewer 2, Comment 3: 3. In Fig 6j, the authors concluded that there is not physical interaction between WH1 and WH2 of MORF. However, only a magnified view of a portion of the NMR spectrum is shown in the figure. The entire NMR spectrum should be presented to clearly show no spectrum changes during the titration experiment.

Author's response: as suggested, the full spectra are now shown in Suppl. Fig. 6.

Reviewer 2, Comment 4: 4. Cryo-EM data is not completely convincing for determining the spatial positions of WH1 and WH2 of MORF. The shape (size) of WH1 is similar to that of WH2. However, the authors modeled WH1 on the dyad and WH2 on the linker DNA. The resolution of the final model is low. Therefore, the densities corresponding to the WH1 and WH2 are obscure, and it is not possible to distinguish which densities are WH1 and WH2. I could not understand the rationale for the modeling of WH1 and WH2. In addition, information of linker DNA sequence is lacking. Does the linker DNA contain CG-rich sequence? If so, the design of the linker DNA is the reason of the modeling of WH1 and WH2? Anyway, the authors should perform cryo-EM experiment using the electron microscope with accelerating voltage of 300 kv and reconsider the experimental design, e.g., preparation of sample grid and cross-linking of the complex, etc., to obtain more clear cryo-EM map for the WH1 and WH2.

Author's response: we have been working on the cryo-EM structure of the WHs-NCP complex for almost two years. We have collected three datasets using 300 kV microscope, tried different additives and freezing grids using Vitrobot/Leica/Chameleon, but only a dataset from 200 kV Talos microscope led to the map with a better cryo-EM density of the NCP-WHs complex.

It's an enormously challenging task to obtain structures of proteins with the nucleosome (please see the most recent study from the Cramer lab published in Nat Str Mol Biol in May 2022- the resolution of NCP-histone H1 complexes are 4-11 Å). Furthermore, the linker DNA of the nucleosome is flexible (see representative 3D classes from a dataset collected using 300 kV microscope, the nucleosome-WHs sample grids prepared using Chameleon, Figure above). This, together with a relatively low occupancy of WHs density, precluded us pushing the resolution further. Despite the low resolution, this structure provides a vital information on the interaction of WHs in the context of the chromatin model.

The linker sequence is now included in the methods section. We have also added the following sentences on page 14: "For reconstitution of NCP₁₉₇ we used DNA₁₉₇ in which 147 bp Widom 601 DNA is flanked by two linker DNA fragments. One linker contains three CpGs and another contains one CpG."

We used the same 197bp DNA from our previous cryo-EM study (PDB ID: 7k61). WH1 was modeled to the linker DNA with three CpGs as WH1 binds specifically to CpG, does not bind to AT, and prefers the linker DNA. In contrast, WH2 is positioned at the dyad, as it doesn't need a linker DNA for binding to NCP (binds equally well to NCP without/with the linker DNA) and has no DNA sequence specificity.

Our new in vitro HAT data with native complexes further confirm the selectivity of WH1 toward the linker DNA (Fig. 7a). The presence of the linker DNA increases the HAT activity of the complexes, which is dependent on WH1, but not on WH2.

Reviewer 2, Comments 5 and 6: 5. In Fig 7a, b and Supplementary Fig 6, the authors described that dissimilar pattern of CSPs of titration experiments. However, the pattern of the spectrum changes of G279, G310, S208, S290 are similar. The authors also claimed slow to intermediate exchange regime on the NMR time scale seen upon titration of MORF WH2 revealed that binding of MORF WH2 is stronger than binding of MORF WH1. However, judging from the Supplementary Fig 6, this spectrum showed typical fast to intermediate regime. In addition, the measurements were not conducted at the same concentration ratio, so it should not be compared. The authors should re-perform the NMR titration experiment at the same concentration ratio and should estimate the K_d values from the signal changes.

Furthermore, identification of the binding surface of WH1 and WH2 for DPF-binding should be required for the arguing the different binding mode between WH1/WH2 and DPF. To do so, the authors should perform the inverse titration experiments using labeled WH1/WH2 and unlabeled DPF.

6, In Fig 7a and 7b, histogram of normalized CSPs, as shown in the Figure 2C, should be prepared in the revised manuscript.

Author's response: the regulatory mechanism appears to be more complex than we initially thought. To avoid any premature conclusions and keep the manuscript focused, we have removed these data from Fig. 7.

Reviewer 2, Comment 7: 7, In Fig 7h and 7i.

Fig 7h showed that DNA binding ability of WH2-DPF of MORF was reduced by DPF. Fig 7i showed that either K182A and K167A in WH2 led to increase in binding of WH2-DPF of MORF to DNA. However, Fig 7h was performed using GST-fused WH2-DPF of MORF. It can not be ruled out that the dimer formation of GST reduces the DNA binding ability of WH2-DPF of MORF. To compare under the same conditions, Fig 7h should be performed using untagged WH2-DPF of MORF.

Author's response: EMSAs with untagged WH2-DPF are compared in Fig. 7c-e.

Reviewer 2, Comment 8: 8, The model structure in Fig 7m was not prepared based on experimental evidence. Simultaneous binding of WH1 and WH2 to DPF should be shown at least.

Author's response: the following sentences have been added to the Fig. 7i legend: MORF_{WH1-WH2}-DPF was first generated using alpha-fold and then aligned with the structure of H3K14cr-bound MORF_{DPF} (PDB ID: 6OIE) and MORF_{WH1-WH2} as in Figure 6m. H3K14cr peptide (dark red) and K167 (green sticks) are labeled.

[Minor comments]

Cryo-EM scFV was used for preparation of the Cryo-EM sample. It binds to the disk of the nucleosome, however, the possibility that binding of scFV inhibit the interaction between nucleosome and WH1-WH2 of MORF can not be excluded. EMSA should be performed for detecting the interaction between nucleosome and WH1-WH2 of MORF in the presence of scFV. – EMSA (Fig. on the left) shows binding of MORF WH1-WH2 to NCP either in the presence or absence of scFV.

Reviewer 3, Comment 1: This reviewer only focused on the DNA methylation analyses and epigenomic sequencing aspects of the manuscript.

The DNA methylation and histone modification sequencing of the manuscript seem ok, although I could find out how exactly the authors performed CpG island recovery assays for unmethylated CpGs (CIRA). They cited one of their earlier paper, but that paper also did not provide detailed protocol. Both referred to the Unmethyl Collector kit (Active Motif) for the CIRA assay, but I could not find any information on this kit and according to the authors' previous paper, the product was discontinued. So how can the authors still perform this assay or could they still perform this assay in the future? I think it would be good for the authors to provide details protocol and cite the original paper for the CIRA assay, so other researchers could reproduce their results if needed.

Author's response: as suggested, we now provide the detailed protocol in the methods section (page 24). Alternatively, MethylCollector MBD capture kit from the same Active Motif can be used followed by CG sequence analysis to distinguish unmethylated and unmethylated CpGs.

REVIEWER COMMENTS

Reviewer #1 (Remarks to the Author):

The authors have addressed my previous comments, and the revised manuscript is improved, and recommended for publication.

Reviewer #2 (Remarks to the Author):

The WH1 consists of 86 amino acid residues, which is suitable for solution NMR. Revealing of the molecular mechanism by which WH1 specifically binds to unmethylated CpG DNA is key point for this manuscript. The best way to know the molecular mechanism for the recognition of the DNA by WH1 is to determine the complex structure at atomic resolution by solution NMR. At least, the author should perform the NMR titration experiment using labeled WH1 and unmethylated CpG DNA and should show the possible molecular mechanism for recognition of unmethylated CpG DNA by WH1; for example, which helix, strand or loop is inserted into the DNA major groove? Which Arg residues are responsible for the recognition of guanine base in the CpG sequence? This structural information may help the construction of the model structure derived from Cryo-EM single particle analysis.

The model of MORF WH1-WH2:NCP complex (Fig 6k and 6I) constructed from Cryo-EM map is unconvincing. Why the authors put WH1 in that on the linker DNA? I couldn't see the EM-map corresponding to the WH1 on the linker DNA because most of the WH1 showing the cartoon model is out of the cryo-EM map judging from Fig 6k and 6I. At least, the authors should have showed the magnified figure of Cryo-EM map around the WH1 and WH2 to help reviewers/readers understand the validity of the model. In addition, WH1 seems to be positioned at the minor groove of the linker DNA. It is curious because WH1 probably recognizes CG sequence from the major groove. Although the EM-map is low resolution, it should be possible to locate the three CpG sequences in the linker DNA. Thus, the authors should present the location of the three CpG sequence in Fig 6k and 6I. Overall, however, the quality of the EM data is too low to discuss the proper location of the WH1 of MOF on the nucleosome. If the authors use Cryo-EM data, cross-link experiments using such as glutaraldehyde and formaldehyde should be performed before preparation of the measurement grid to prepare the stable complex.

This model in Fig 7i is not still convincing. In the first manuscript, this model is supported by the NMR data that show the WH1 and WH2 interact with DPF using different binding surfaces of DPF, but this NMR data is removed in the current version. In addition, AF2 generally generates five independent structures. Which model was used for the model construction? The authors should display the 'Expected position error' (attached figure) to validity the spatial arrangement between WH1, WH2 and DPF.

Minor point

In Fig 3h, cartoon presentation with the side chains of key residues and pLDDT estimated by AF2 also should be shown.

Editorial note: 'Expected position error' attached figure from Reviewer 2

Reviewer #3 (Remarks to the Author):

The authors have answered my question in the revision.

We would like to thank Reviewers for the insightful and very constructive comments, which were helpful in revising and strengthening this manuscript.

Reviewer 2, Comment 1: The WH1 consists of 86 amino acid residues, which is suitable for solution NMR. Revealing of the molecular mechanism by which WH1 specifically binds to unmethylated CpG DNA is key point for this manuscript. The best way to know the molecular mechanism for the recognition of the DNA by WH1 is to determine the complex structure at atomic resolution by solution NMR. At least, the author should perform the NMR titration experiment using labeled WH1 and unmethylated CpG DNA and should show the possible molecular mechanism for recognition of unmethylated CpG DNA by WH1; for example, which helix, strand or loop is inserted into the DNA major groove? Which Arg residues are responsible for the recognition of guanine base in the CpG sequence? This structural information may help the construction of the model structure derived from Cryo-EM single particle analysis.

Author's response: as suggested, we have performed NMR titration experiments using ¹⁵N-labeled WH1 and unmethylated CpG DNA. While WT WH1 binds to CpG-DNA (new Fig. 3l), mutations in the α 1 helix (KKK) or mutations in the loops connecting α 1 and α 2 and β -strands (KRK) disrupt this binding, indicating that α 1 and the loops are involved in the interaction with CpG-DNA (new Fig. 3m, n and Suppl. Fig. 5).

In support of the NMR data, fluorescence polarization assays (new Fig. 7k) reveal that these KKK and KRK mutants of WH1 are impaired in binding to the nucleosome.

To show *in vivo* the importance of the DNA-binding α 1 and the loops of WH1 for localization to CpG, we have performed ChIP qPCR assays (new Fig. 5d). The KKK and KRK mutants of WH1 lost the ability to localize to the genomic sites. The new ChIP-seq analysis of WH1 of MORF confirms that recognition of the CpG sequence is conserved in MORF (Fig. 5b, c).

To further characterize the DNA-binding mechanism of WH1, we carried out analysis of the crystal structures of DNA-bound WHs. It reveals high similarity of the MORF/MOZ WH1 sequence to that of WH of SAMD1, an atypical WH that inserts α 1 and the loop connecting α 1 and α 2 in the CpG-containing major groove of DNA, while inserting its atypically long β -hairpin into a neighboring minor groove. We now compare these atypical WHs on pages 9-10. The AF model of MORF_{WH1} overlays very well with the structure of SAMD1 WH (rmsd of 0.6 Å), suggesting that the DNA-binding mechanism is conserved (new Fig. 3i and 8c, d).

Reviewer 2, Comment 2: The model of MORF WH1-WH2:NCP complex (Fig 6k and 6l) constructed from Cryo-EM map is unconvincing. Why the authors put WH1 in that on the linker DNA? I couldn't see the EM-map corresponding to the WH1 on the linker DNA because most of the WH1 showing the cartoon model is out of the cryo-EM map judging from Fig 6k and 6l. At least, the authors should have showed the magnified figure of Cryo-EM map around the WH1 and WH2 to help reviewers/readers understand the validity of the model.

Author's response: as suggested, the magnified figure of the cryo-EM map around WH1 and WH2 is now shown in Fig. 8b.

We have clarified in concluding remarks and throughout the text that WH1 belongs to a subset of atypical WHs with a long β -hairpin. The DNA binding mechanism of this subset of WHs is different from the DNA binding mechanism of typical WHs, such as WH2. WH1 does not contain positively charged residues in the α 3 helix, necessary for binding to the nucleosome dyad and

the long β -hairpin precludes binding there due to steric clashes. WH1 binds exclusively to CpG, hence the CpG-linker NCP₁₉₇ was designed to monitor this interaction.

To further validate the NCP binding mechanism, we have carried out

- FRET experiments to confirm binding of WH2 to the dyad of the nucleosome that leads to an increase in the DNA wrapping and stabilization of the nucleosome, also seen due to binding of H1 at the dyad (new Fig. 8f, g and Suppl. Fig. 10).
- EMSA assays to confirm that the KKK and KRK mutants of WH1 do not bind to the CpG-NCP₁₉₇ (new Fig. 8e and Suppl. Fig. 9). These results are in agreement with NMR titration data shown in Fig. 3l-n.

In addition, WH1 seems to be positioned at the minor groove of the linker DNA. It is curious because WH1 probably recognizes CG sequence from the major groove. – the atypical WHs insert a long β -hairpin into the minor groove.

Although the EM-map is low resolution, it should be possible to locate the three CpG sequences in the linker DNA. Thus, the authors should present the location of the three CpG sequence in Fig 6k and 6l. – as suggested, the CpG sequences are colored red and labeled in Fig. 8a, b.

Overall, however, the quality of the EM data is too low to discuss the proper location of the WH1 of MOF on the nucleosome. If the authors use Cryo-EM data, cross-link experiments using such as glutaraldehyde and formaldehyde should be performed before preparation of the measurement grid to prepare the stable complex. – in the revised Fig. 8 a,b, WH1 is no longer docked into the density. We also used formaldehyde for crosslinking, however the screen images didn't show uniform particles.

Reviewer 2, Comment 3: This model in Fig 7i is not still convincing. In the first manuscript, this model is supported by the NMR data that show the WH1 and WH2 interact with DPF using different binding surfaces of DPF, but this NMR data is removed in the current version. In addition, AF2 generally generates five independent structures. Which model was used for the model construction? The authors should display the 'Expected position error' (attached figure) to validity the spatial arrangement between WH1, WH2 and DPF.

Author's response: two NMR titration panels of ¹⁵N-labeled DPF were removed because we cannot confirm the direct interaction with DPF by reciprocal NMR titration of unlabeled DPF (Fig. on the left). Because all three domains bind DNA, CSPs initially observed are likely due to DNA that was not fully removed during purification of these domains. We have revised Fig. 9i to depict a schematic of confirmed direct interactions.

The AF model of MORF is taken from UniProt (Q8WYB5). There is only one AF model in UniProt.

Minor point

In Fig 3h, cartoon presentation with the side chains of key residues and pLDDT estimated by AF2 also should be shown. – the side chains of the key residues are now shown in Fig. 3i. The AF model of WH1 of MORF is taken from UniProt (Q8WYB5).

REVIEWERS' COMMENTS

Reviewer #2 (Remarks to the Author):

The authors have performed additional experiments appropriately and addressed my concerns sincerely .

The manuscript and figures have been suitably modified.

I hope my review was helpful in improving the manuscript.